

# Consequences of the Aral Sea restoration for its present physical state: temperature, mixing, and oxygen regime

Georgiy B. Kirillin[1], Tom Shatwell[2], and Alexander S. Izhitskiy[3]

[1]Leibniz-Institute of Freshwater Ecology and Inland Fisheries (IGB), Berlin, Germany
[2]Ostwestfalen-Lippe University of Applied Sciences and Arts, Germany
[3]Shirshov Institute of Oceanology, Russian Academy of Sciences, Moscow, Russia

**Correspondence:** Georgiy Kirillin (georgiy.kirillin@igb-berlin.de)

**Abstract.** The restoration of the North Aral was an unprecedented effort to save a large water basin by construction of a dam that separates it from the rest of the desiccating Aral Sea area. As a result, the lake volume has stabilized at 27.5 $\mathrm{km}^3$, the area has increased from 2800 $\mathrm{km}^2$ (2006) to 3400 $\mathrm{km}^2$ (2020), and the salinity has dropped from 18 to 10 $\mathrm{g\,kg}^{-1}$. The consequences of this unique experiment include highly dynamic changes of the thermal conditions, seasonal stratification, ice
regime, and dissolved oxygen content and remain not fully quantified to date. We analyze the current state of the North Aral Sea with regard to stabilization of its long term dynamics, as well as consider the possible future projections in view of the global change effects on the regional hydrological regime and potential water management measures. Using data from a series of expeditions to the North Aral Sea in 2016-2019 and year-long continuous monitoring of the annual thermal and oxygen regime by an autonomous mooring station, we present the first comprehensive analysis of the North Aral system behavior on
seasonal to interannual scales after its "cold restart". We demonstrate that the new seasonal mixing regime is intermediate between dimictic and polymictic, with relatively weak summer thermal stratification occupying only a small deep part of the lake. Salinity does not contribute to the summer density stratification but a stable salinity stratification can develop during ice melt in late winter. On the background of weak thermal stratification, highly energetic internal waves with periods of ~4.5 days dominate the near-bottom dynamics and facilitate mixing at the lake bottom. As a result, the bulk of the water column remains
well saturated with oxygen throughout the year. However, low-oxygen conditions may develop in the deepest part of the lake in mid-summer. In summary, the mixing regime of the restarted lake favors vertical transport of dissolved matter and water-sediment mass exchange ensuring oxygenation of deep waters and supply of nutrients to the upper water column. While the North Aral Sea is restored to the well-mixed state similar to that before its desiccation started, its seasonal mixing regime is currently in unstable equilibrium, wobbling between polymictic and dimictic conditions. The fragility of this seasonal pattern
is demonstrated by modeling results: slight changes of the water level or transparency may turn the Aral Sea to steadily dimictic or polymictic state.

## 1  Introduction

Drying of lakes as a result of water resources mismanagement and climate change is a growing environmental concern around the world. Large endorheic lakes are particularly vulnerable to desiccation. Many of these lakes, such as the Aral Sea, Lake



Chad, Lake Urmia, Great Salt Lake, and the Dead Sea are experiencing acute shrinkage of their water volume, threatening large-scale environmental, ecological, and socio-economical consequences. As the problem of drying lakes grows globally, restoration efforts are becoming increasingly important. Various restoration measures have been proposed, such as regulating water withdrawal, water saving techniques in agriculture and industry, and mitigation of climate change through global cooperation and policy making. A particularly effective approach consists in lake level replenishment by managing the major inflows.

Some notable examples include: an ambitious proposal involving transfer of water from the Congo River Basin to Lake Chad (Adeniran and Daniell, 2021; Kitoto, 2021), which has shrunk by as much as 90 % since the 1960s, primarily due to overuse of the water for irrigation and the effects of climate change; the Dead Sea preservation project which aims to stop the decline of the Dead Sea (Asmar and Ergenzinger, 2002; Asmar, 2003), considering the potential of a canal from the Red Sea to the Dead Sea to stabilize water levels; the Lake Urmia restoration program (Danesh-Yazdi and Ataie-Ashtiani, 2019; Parsinejad

et al., 2022), which includes diverting water from nearby rivers and a plan to release water from upstream dams when levels are critically low. Such "megaprojects" have multiple, barely predictable consequences for lake ecosystems and the regional water budget. In this regard, the outcomes of the Aral Sea restoration project are particularly insightful for the current and future projects, serving as an example of a successful large scale experiment with multifaceted consequences for the lake and its environment.

Once the fourth largest lake in the world, the Aral Sea has been shrinking since the 1960s after the rivers that fed it were diverted by Soviet irrigation projects. Rapid desiccation of the Aral Sea draws continuous attention of researchers as an example of fast anthropogenically driven change of a large aquatic ecosystem at unprecedented spatial scales. The formerly brackish dimictic lake was reduced to several (semi-)isolated water bodies with significantly divergent hydrological and biogeochemical conditions (Izhitskiy et al., 2016). By the 2000s, due to the active use of the two main tributaries, Amu Darya and Syr Darya,

the lake had shrunk within 40 years to 10 % of its original surface area (Zavialov, 2007), resulting in severe ecological consequences, loss of fisheries, and local climate extremes. At the same time, the water salinity in isolated parts of the Aral Sea increased from $11\,\mathrm{g\,kg^{-1}}$ to more than $200\,\mathrm{g\,kg^{-1}}$ (Andrulionis et al., 2021; Izhitskiy et al., 2021; Andrulionis et al., 2022).

As a countermeasure to prevent further desiccation, a dam was constructed in 2005 separating the northern part of the Aral Sea from the rest of the former lake basin (Figure 1). The effort stabilized the volume and salinity and was widely recognized

as an exceptional success in large scale water management and restoration (Micklin, 2014). The 12 km long Kokaral Dam confined the discharge of the Syr Darya River to the North Aral Sea, stabilized its water level at 42.5 m a.s.l. (compared to about 40.5 m above sea level before dam construction), and reduced the average salt content from $18\,\mathrm{g\,kg^{-1}}$ in August 2002 (Friedrich and Oberhänsli, 2004; Friedrich, 2009) to $11\,\mathrm{g\,kg^{-1}}$ in October 2014 (Izhitskiy et al., 2016), the latter value being comparable to that before the desiccation (Bortnik and Chistyaeva, 1990). Several studies described significant changes in the

biological characteristics of the lake during the period after its isolation (Plotnikov et al., 2016; Massakbayeva et al., 2020). Water level and salinity fluctuations led to diversity changes in plankton and zoobenthos communities of the North Aral Sea before (Aladin et al., 2005) and after the Kokaral Dam construction (Krupa et al., 2019; Klimaszyk et al., 2022) along with changes in ichthyofauna and fisheries (Ermakhanov et al., 2012). While rapid changes in fish and zooplankton communities



were documented during the 15 years of restoration, little was known about the seasonal mixing regime of the restored Aral
Sea, including thermal and salinity structure of the water column.

The seasonal mixing regime is a fundamental lake characteristic, which governs the distribution of nutrients, oxygen, and organisms throughout the water column and determines the ecological balance and overall health of a lake (Hutchinson, 1957). By this, seasonal mixing regulates the primary production and sustains biodiversity of aquatic life. A crucial role of seasonal mixing consists in oxygenating the lake water column (Golosov et al., 2007; Valerio et al., 2019; Pilla et al., 2023). Oxygen
from the surface gets mixed down to deeper waters, providing essential life support for aerobic organisms, including fish and bacteria. Conversely, it also helps to release gases generated at the bottom of the lake, such as hydrogen sulfide and methane and prevents toxic gas accumulation.

The far-reaching implications of seasonal mixing disruption for the health and biodiversity of lake ecosystems have been demonstrated by the fate of the southern part of the Aral Sea, which was not affected by the restoration measures (Izhitskaya
et al., 2019; Izhitskiy et al., 2021): after seasonal mixing was canceled there by strong vertical salinity gradients, the residual Aral waters have partially turned into anoxic environments with extremely low biodiversity and high rates of methane production in deep waters.

Before desiccation, the Aral Sea was a typical *dimictic* lake, with two events of full mixing (spring and fall "overturns") separating the winter and summer stagnation periods (Zavialov, 2007). The North Aral was ice-covered from late December or
early January for 120-140 days. Summer was characterized by strong thermal stratification with surface temperatures >20 °C and bottom temperatures slightly above the maximum density value of ∼1.6 °C. Salinity varied seasonally between 9.0 and 10.8 g kg$^{-1}$ and was nearly evenly distributed vertically by spring and fall overturns.

Almost uniform vertical temperature profiles were observed in late summer 2002 before the Kokaral Dam construction (Friedrich and Oberhänsli, 2004) and in autumn 2014 after the restoration (Izhitskiy et al., 2016) suggesting that the lake mixes
down to the bottom at least once a year. However, before restoration, hypoxic conditions were reported near the lake bottom (Friedrich and Oberhänsli, 2004; Friedrich, 2009) indicating active oxygen consumption in the deep thermally stratified part of the water column during the summer stagnation period. The seasonal mixing and stratification regime of the North Aral Sea after restoration remained unknown, but is crucial to understanding the consequences of the restoration measures for the lake ecosystem at all trophic levels. Located in the arid continental climate, the North Aral undergoes strong seasonal variations in
the heat exchange with the atmosphere and, as a result, in the water temperatures and density stratification. The lake surface remains isolated from the atmosphere by the ice cover during winter months (Kouraev et al., 2004), which leads to winter hypoxia development in lakes (Golosov et al., 2007). In summer, in turn, deep hypoxia can be accelerated by the strong surface heating and development of thermal stratification, preventing oxygen exchange between surface and bottom waters. The North Aral ecosystem remains one of the largest ecosystems in the arid region of Central Asia. Taking into account its vulnerability
to water use and climatic changes in hydrological balance in the second largest endorheic lake basin (after the Caspian Sea), understanding the response of the lake to mitigation measures is crucial for planning future activities in the context of climate-driven water scarcity and growing irrigation.



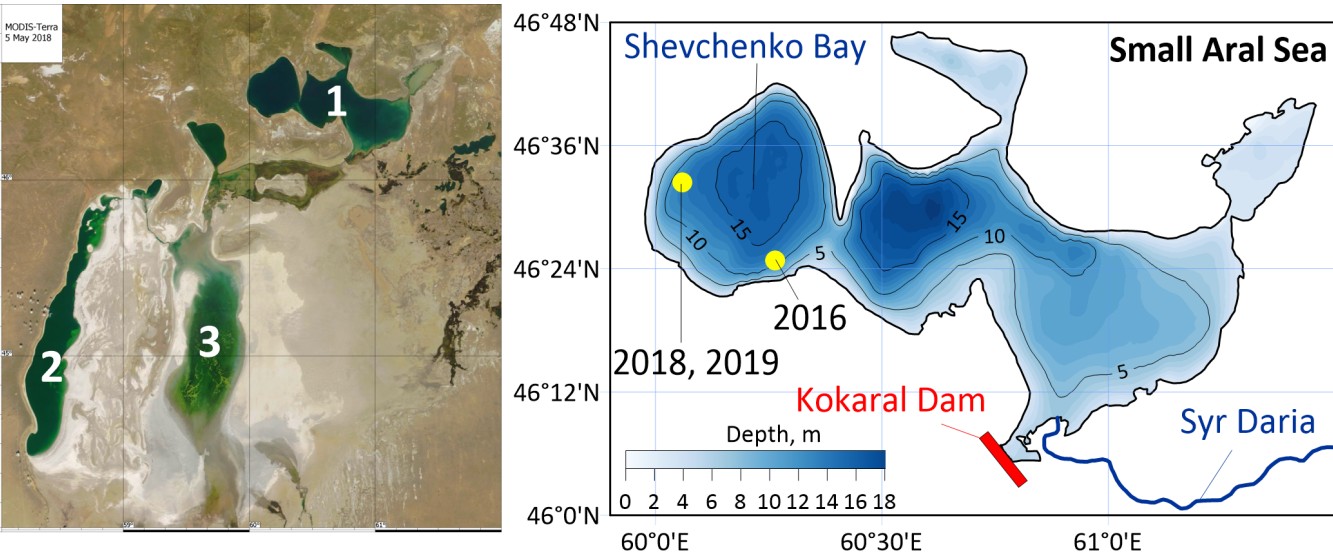

**Figure 1.** Residual basins of the Aral Sea by MODIS Terra from May 5, 2018: 1 – the North (Small) Aral Sea, 2 – the Western South (Large) Aral Sea, 3 – the Eastern South (Large) Aral Sea; approximate bathymetry map of the North Aral Sea with location of measurement sites in 2016 and 2018-2019.

Summarizing the information above, the following basic hypotheses can be formulated on the present and future mixing and oxygen dynamics of the North Aral Sea: (i) The lake may have re-established a dimictic pattern, with two mixing events in spring and fall, separated by periods of stagnation in winter and summer; (ii) Hypoxic conditions could develop in the lake's deep areas during stagnation phases, due to limited deep oxygen replenishment; and (iii) gradual shifts in water level may cause the lake to oscillate between poly- and dimictic mixing regimes, depending on changes in lake depth. Below, we use data from several measurement campaigns, including year-round monitoring of temperature and near-bottom oxygen concentrations, as well as modeling, climate scenarios, and remote sensing data, aiming at

– quantifying the consequences of large scale lake restoration measures for the annual thermal and mixing regime of the North Aral, including potential stagnation;

– revealing the effects of the mixing regime on dissolved oxygen levels in deep waters;

– assessing possible future impacts of regional hydrological regime changes on lake mixing.





## 2   Materials and Methods

### 2.1   Sampling sites and procedures

We performed our observational studies in the western part of the North Aral Sea, the Shevchenko Bay (Fig. 1), during 3 short term intense field surveys that took place between 2016 and 2019. On 24 June 2016, direct measurements were carried out in the southern part of Shevchenko Bay at coordinates 46.414° N, 60.265° E and water depth of 13.1 m. On 24 September 2018 and 28 September 2019 direct measurements were carried out in the western part of Shevchenko Bay at coordinates

46.54027° N, 60.05688° E and water depth of 11.9 m. During each of the surveys, *in situ* measurements included vertical profiling of the main physical and chemical characteristics and water sampling. Profiles of conductivity, temperature, depth, fluorescence and dissolved oxygen were taken in June 2016 and September 2018 with a Rinko CTD Profiler (JFE Advantech, Japan). In September 2019, profiles of conductivity, temperature and depth were taken with a RBRConcerto CTD (RBR Ltd., Canada). The water samples were taken with a 5 l Niskin Hydro-Bios bottle and analyzed later in the laboratory for salinity

and ionic compositions using the methods described by Andrulionis et al. (2022)

### 2.2   Long term autonomous monitoring

An autonomous moored chain was installed in Shevchenko Bay at the coordinates 46.54° N, 60.06° E and a water depth of 11.8 m, just near the sampling site. The chain was deployed on 24 September 2018 and recovered on 28 September 2019. The moored chain was equipped with 9 temperature sensors TR-1060 (RBR Canada, accuracy 0.002 K) distributed between the

surface and the bottom of the water column at depths of 2.22, 2.77, 3.42, 4.82, 5.82, 6.92, 8.92, 10.02 and 11.77 m. In addition, a dissolved oxygen (DO) logger D-Opto (ZebraTech, New Zealand, accuracy 1 % of measurement or 0.02 ppm, whichever is greater) was deployed at 10.02 m water depth. *In situ* measurements were performed with sampling rates of 30 seconds for water temperature and 1 hour for dissolved oxygen concentration.

### 2.3   Framework of stratification analysis

We analyzed thermal stratification in terms of the buoyancy frequency

$$N(z) = \sqrt{\frac{g}{\rho}\frac{\partial \rho}{\partial z}}, \tag{1}$$

where $g$ is the gravity acceleration, $\rho$ is the water density, calculated as a function of temperature and salinity. $\rho_0$ is the reference water density in the Boussinesq sense, which can be taken as the mean density constant in time and depth across the water column.

For estimation of vertical density gradients and $N(z)$ we used the TEOS-10 toolbox (McDougall and Barker, 2011). The salt composition of the North Aral water (Andrulionis et al., 2022) (see also Table 1 in Results) is slightly different from that of the seawater (Millero et al., 2008), however, the water column is well mixed with respect to salinity, making the contribution of salinity to vertical density stratification nearly negligible.




Density stratification estimates were further applied to evaluate characteristics of internal gravity waves. For a water column of depth $H$, the buoyancy frequency determines the celerity (phase speed) of a long internal wave,

$$C_m = \frac{1}{m\pi} \int\limits_{-H}^{0} N(z)dz, \tag{2}$$

where $m = 1, 2, 3 \ldots$ is the vertical "natural wave mode", and $m = 1$ corresponds to the gravest (slowest) wave $C_1$ (Gill, 1982). For the two-layered vertical stratification with a sharp density gradient separating a lighter homogeneous layer (epilimnion) from the denser hypolimnion beneath, typical for seasonally stratified enclosed lakes, the expression for the first-order internal wave speed $C_1$ reduces to

$$C_1 = (g' h_{eq})^{1/2} = \sqrt{g \frac{\rho_2 - \rho_1}{\rho_0} \frac{h_1 h_2}{h_1 + h_2}}. \tag{3}$$

Here, $g'$ is the "reduced gravity" expressing the density jump across the infinitesimal boundary between the two layers ($\rho_2 > \rho_1$), and $h_{eq}$ is the "equivalent depth". The ratio of $C_1$ to the Coriolis acceleration $f$ represents the first-order baroclinic Rossby radius, a fundamental length scale expressing the effect of earth rotation on the internal wave characteristics. Here, $f = 2\omega \sin\phi \approx 10^{-4}$ s$^{-1}$ is the Coriolis frequency, $\omega = 7.29 \cdot 10^{-5}$ rad·s$^{-1}$ is the angular speed of earth rotation, $\phi$ is the geographic latitude ($\sim 46.5°$ N for the North Aral Sea).

The dimensionless ratio of the baroclinic Rossby radius to the lateral scale of the lake $L$ is the Burger number:

$$S = \frac{C_1}{fL}. \tag{4}$$

As a characteristic horizontal scale $L$ of the basin-scale motions, the average radius of a circle approximating the Shevchenko Bay (Fig. 1) was chosen. The reason for this choice is isolation of the bay from the eastern part of the lake constraining the large scale water motions within its nearly round-shaped basin. The model of basin scale internal waves in a large circular lake (Csanady, 1967; Lamb, 1932) was applied in the further analysis of these motions. According to the model solution of the water motion equations, the internal waves are characterized by a set of frequencies $\omega$, determined by two sets of eigenvalue problems:

$$(\sqrt{\sigma^2 - 1}/S)J_{n-1}(\sqrt{\sigma^2 - 1}/S) + n(1/\sigma - 1)J_n(\sqrt{\sigma^2 - 1}/S) = 0 \quad \text{at} \quad \sigma > 1 \tag{5}$$

$$(\sqrt{1 - \sigma^2}/S)I_{n-1}(\sqrt{1 - \sigma^2}/S) + n(1/\sigma - 1)I_n(\sqrt{1 - \sigma^2}/S) = 0 \quad \text{at} \quad -1 < \sigma < 0 \tag{6}$$

Here, $n = 1, 2, ...$ is the radial wave mode, $S$ is the Burger number (Eq. 4), $J_n$ and $I_n$ are the Bessel functions and the modified Bessel functions of the first kind of order $n$, respectively, and $\sigma = \omega/f$ is the dimensionless wave frequency. The solution of Eqs. (5)-(6) gives a pair of frequencies, one corresponding to a cyclonic ($\sigma < 0$) Kelvin wave with amplitudes growing towards the lake shores due to the exponential form of $I_n$, the other corresponding to an anticyclonic ($\sigma > 1$) Poincaré wave with amplitudes varying periodically across the lake due to the periodic form of $J_n$. For a real lake, the wave frequencies can be estimated from data on vertical density stratification and the lake size, and may be identified in the spectra of density (temperature) oscillations.





To estimate the stratification stability on the lakewide scales we utilized the Schmidt stability parameter

$$St = \rho_0 g \bar{h} (Z_v - Z_g), \qquad [\mathrm{kg\,s^{-2}}] \tag{7}$$

where

$$Z_v = V^{-1} \int\limits_{-H}^{0} A(z) z \, \mathrm{d}z \quad \text{and} \quad Z_g = M^{-1} \int\limits_{-H}^{0} A(z)\rho(z) z \, \mathrm{d}z$$

are the heights of the lake center of volume and that of gravity, respectively; $M$ is the lake mass; $V$ is the lake volume; $\rho_0$ is the mean lake density; $\bar{h}$ is the mean lake depth. Following from the definition, $St$ is a measure of the potential energy excess created by vertical density stratification. Accordingly, $St$ can be directly compared to the kinetic energy input from wind and(or) buoyancy loss at the lake surface due to heat exchange with the atmosphere. In particular, the ratio of the surface wind stress $\tau$ to $St/h_{eq}$ is an approximate measure of the isopycnals slope due to the wind energy input at the lake surface, and may be used to estimate the probability of lake overturn by wind mixing.

The spectra of temperature oscillations were estimated by the Welch method: splitting of the original temperature time series into several 50 % overlapping segments with subsequent tapering of the segments using a Hamming window to reduce the cut-off effects of the limited data length, calculation of the discrete Fourier transform and averaging the resulting power spectral densities over all segments. The segment length was chosen as 40 days, being significantly shorter than the period of seasonal variations but exceeding the typical synoptic time scales of 5 to 10 days.

## 2.4 Modelling

We applied the lake model FLake (Mironov et al., 2010; Kirillin et al., 2011) for simulation of the lake temperature cycle, water mixing, and seasonal ice cover. FLake is a one-dimensional model using a multilayer parametric representation of the vertical temperature structure in the ice-water-sediment system combining integration of the heat budget in each layer with semi-empirical knowledge on the vertical thermal structure. The upper water layer is modeled following the mixed boundary layer approach (Kraus and Turner, 1967; Kraus, 1972), while temperature profiles in the stably stratified part of the water column (hypolimnion), within the ice cover, and in the thermally active upper layer of lake sediments are parameterized using self-similar time-dependent functions of the vertical coordinate. The hypothesis of self-similarity (Kitaigorodski and Miropolski, 1970; Barenblatt, 1978; Zilitinkevich et al., 1979) applied to vertical temperature distributions in layers with quasi-homogeneous physical conditions reduces the modeling approach to solving a set of ordinary differential equations in the time domain instead of discretizing the original partial differential equations along the vertical coordinate. The solution of the initial value problem for the set of three ODEs is performed using the Newton method with a constant time step. This approach distinguishes FLake from other lake temperature models and provides it with high computational efficiency (Kirillin et al., 2011; Thiery et al., 2014b) allowing flexible applications in long-term scenarios of lake response to climate change or to the anthropogenic change of external fluxes. FLake has proven its robustness and reliability for simulating long-term temperature and mixing conditions in various freshwater lakes and reservoirs (Kirillin et al., 2011; Thiery et al., 2014a; Kirillin et al., 2017; Su et al., 2019; Almeida et al., 2022).





*Salinity extension*. To account for the effects of dissolved salts on vertical mixing and ice formation in the brackish waters of the North Aral Sea, FLake was extended to include salinity effects on the thermal conditions. The contribution of salinity to the vertical density stratification in brackish lakes is typically minor compared to that of the vertical temperature gradients: since the temperature of the maximum density $T_{md}$ of brackish waters is above the freezing temperature, the lakes are effectively

vertically mixed with respect to salinity by convection when surface temperatures cross $T_{md}$ due to seasonal surface cooling. The salt fluxes by freshwater discharge into the lake, evaporation, and cryoconcentration (salt exclusion due to ice formation) are usually too low to produce significant stratification, although freshwater release from melting ice may affect the upward heat transport from water to the ice base and decelerate the ice thaw. Therefore, salinity was taken as constant in time and depth, i.e. was assumed to be evenly distributed vertically, and the salt fluxes from(to) the atmosphere, ice, sediment, and tributaries were

neglected. However, salinity significantly affects temperature and ice regime by changing the maximum density temperature, thermal expansion coefficient and the freezing point. In the current version of FLake, the three properties were parameterized as functions of water salinity. The assumption allowed the major salinity effects to be incorporated in the bulk of brackish lakes without increasing the model complexity.

The following parameterizations of salinity effects were applied. The original quadratic equation of state of the fresh water

was adopted,

$$\rho_w = \rho_r \left[ 1 - \frac{1}{2} a_T \left( T - T_{md0} \right)^2 \right], \tag{8}$$

where $\rho_w$ is the water density, $\rho_r = 999.98 \approx 1.0 \cdot 10^3$ kg·m$^{-3}$ is the maximum density of the fresh water at the temperature $T_{md0} = 277.13$ K, and $a_T = 1.6509 \cdot 10^{-5}$ K$^{-2}$ is an empirical coefficient (Farmer and Carmack, 1981). Equation (8) is the simplest equation of state that accounts for the fact that the temperature of maximum density of the fresh water exceeds

its freezing point $T_{f0} = 273.15$ K. According to Eq. (8), the thermal expansion coefficient $\alpha_T$ and the buoyancy parameter $\beta$ depend on the water temperature, $\beta(T) = g\alpha_T(T) = ga_T(T - T_{md}(S))$, where $g = 9.81$ m·s$^{-2}$ is the acceleration due to gravity. To incorporate the salinity effects on water density, the temperature of maximum density $T_{md}$ was represented by a function of salinity $S$ [g kg$^{-1}$ or ‰] following Caldwell (1978)

$$T_{md}(S) = T_{md0} - 0.2229S. \tag{9}$$

The salinity effect on the thermal expansion coefficient $\alpha_T$ was parameterized by simplified polynomials of Bryden (1973) with coefficients adjusted according to Caldwell (1978),

$$\alpha_T(T,S) = \left[ -56.6537 + 3.3858S - 0.81218 \cdot 10^{-2} S^2 + 14.2407(T - T_{f0}) \right] 10^{-6}. \tag{10}$$

The melting temperature was calculated from water salinity after Feistel and Hagen (1998)

$$T_f(S) = T_{f0} + S(-57.5 + 1.710523\sqrt{S} - 0.2154996S)10^{-3}.$$

*Model setup*. Flake was forced with meteorology from the ERA5 reanalysis (Hersbach et al., 2020), including short and long wave radiation, 10 m wind speed, air temperature, and humidity. The lake-specific parameters were modelled on Shevchenko





**Table 1.** Salinity and major ions in the North Aral Sea waters in $\mathrm{g\,kg^{-1}}$. Data for August 2002 as a pre-Kokaral conditions are given from Friedrich and Oberhänsli (2004) for comparison. Values for September 2019 are taken from Andrulionis et al. (2022)

|  | Depth | Salinity | $Cl^-$ | $SO_4^{2-}$ | $HCO_3^-$ | $Na^+$ | $K^+$ | $Ca^{2+}$ | $Mg^{2+}$ |
|---|---|---|---|---|---|---|---|---|---|
| August, 2002 | 0 m | 17.54 | 5.98 | 6.11 | 0.21 | 3.54 | 0.19 | 0.53 | 0.96 |
|  | 10 m | 16.72 | 5.98 | 6.13 | 0.22 | 2.94 | 0.17 | 0.45 | 0.82 |
| September, 2019 | 0 m | 10.68 | 2.84 | 3.90 | 0.17 | 2.17 | 0.12 | 0.50 | 0.50 |
|  | 12.4 m | 10.48 | 2.85 | 3.76 | 0.17 | 2.84 | 0.10 | 0.49 | 0.26 |

Bay, with the lake depth of 11 m, salinity of 11 ‰, characteristic fetch of 53 km, and light extinction coefficient of 0.5 $\mathrm{m^{-1}}$ corresponding to a Secchi depth of 3.5 m using the equation of Poole and Atkins (1929). The simulation period was from 01 September 2018 to 31 December 2020, with a 9 month spinup period and 1 hour timestep. The initial profile on Jan 1 was set with a surface and bottom temperature of 0 and 4 °C. Stratification was inferred when the surface-bottom temperature difference exceeded 1 °C, and the length of the longest uninterrupted period was considered its seasonal duration. Model performance was assessed with root mean square error ($RMSE$) of surface and bottom temperatures as:

$$RMSE = \sqrt{\frac{\sum_{i=1}^{n}(o_i - m_i)^2}{n}} \tag{11}$$

where $o_i$ and $m_i$ are observed and modeled values, respectively.

## 3  Results

### 3.1  Salt composition and vertical distributions of major water characteristics by results of 2016-2019 surveys

According to the observational results, the North Aral Sea is currently a brackish lake without pronounced vertical haline stratification in the water column. Salinity of surface waters ranged from 10.71 $\mathrm{g\,kg^{-1}}$ in September 2018 to 10.68 $\mathrm{g\,kg^{-1}}$ in September 2019. The differences between surface and bottom values were less than 0.1 $\mathrm{g\,kg^{-1}}$ in 2018 and about 0.2 $\mathrm{g\,kg^{-1}}$ in 2019. Note that these salinity values refer to the western part of North Aral, which is the farthest from the Syr Daria delta. Salinity is expected to decrease eastwards: In 2015, salinity values as low as 9.94 $\mathrm{g\,kg^{-1}}$ were reported in the mouth area of the Syr Darya River (Andrulionis et al., 2022). Table 1 summarizes the ionic composition of North Aral before and after the construction of the Kokaral dam. The data demonstrate a decrease in salinity by almost 7 $\mathrm{g\,kg^{-1}}$ relative to the pre-Kokaral conditions. The freshening of lake water was accompanied by changes in the ionic composition, mainly associated with an increase in the SO$_4$/Cl ratio (Andrulionis et al., 2022). Throughout the period 2016-2019 the vertical distribution of temperature, conductivity, DO and chlorophyll a was nearly uniform in the upper 12 m of the water column (Figure 2). In summer (see data from June 2016, Fig. 2a), a stable temperature stratification $\sim$1 $\mathrm{K\,m^{-1}}$ develops below 12 m water depth, accompanied by a slight (from 13.7 to 13.1 $\mathrm{mS\,cm^{-1}}$) downward decrease of electrical conductivity (corresponding approximately to a 0.5 $\mathrm{g\,kg^{-1}}$ drop in salinity). The latter can be attributed to the intense evaporation from the lake surface during the preceding

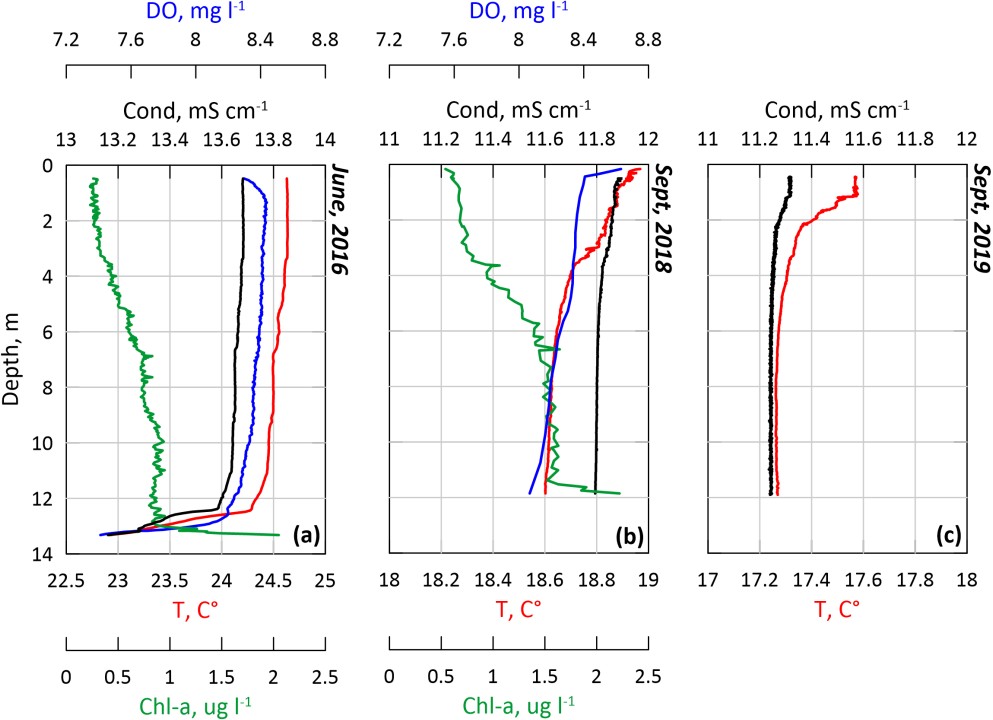

**Figure 2.** Vertical profiles of temperature (red), electrical conductivity (black), dissolved oxygen (blue) and fluorescence (green) observed in the North Aral Sea in June 2016 (a), September 2018 (b), and September 2019 (c).

summer together with the strong mixing across the upper 10 to 12 m of the water column. Fall temperatures do not reveal any strong vertical gradients (0.025 $\mathrm{K\,m^{-1}}$ in 2018 and 0.030 $\mathrm{K\,m^{-1}}$ in 2019, Fig.2bc). The slight increase of temperature values near the surface resulted apparently from the diurnal cycle of surface heating. It is noteworthy that the mean values of electrical conductivity continuously decreased from 13.5 in 2016 to 11.85 in 2018 to 11.3 in 2019. While the records might be affected by seasonal and interannual variability of the freshwater budget, they suggest ongoing freshening of the Aral water. The recorded

Chlorophyll-a values of $\sim 2\,\mu\mathrm{g\,L^{-1}}$ are characteristic of oligo-mesotrophic conditions (Carlson, 1977). The Chl-a values from fluorescence measurements were not not calibrated against standard laboratory methods and may deviate from the real values. Nevertheless, the values are 3 to 10 times below the typical eutrophic Chl-a concentrations suggesting a mesotrophic state of the lake.

### 3.2 Seasonal thermal regime from continuous mooring observations

According to the autonomous measurements, the temperature cycle of the North Aral is characteristic of holomictic lakes (those entirely mixing at least once during the annual cycle) and is distinguished by two periods of vertical thermal stratification separated by periods of nearly homothermal conditions (Fig. 3a). The period of winter stratification coincides with the period of





ice cover and is characterized by the downward temperature increase from the ice base to the sediment surface. In summer, the water column is generally stratified with a temperature drop of up to $\sim 10\,°C$ from surface to bottom, whereas the stratification at the measurement site was periodically interrupted by short term full mixing events. The overall seasonal stratification pattern of the North Aral is thereby between cold monomicitic and dimictic according to the classification of Hutchinson (Hutchinson, 1957). Four periods are distinguishable in the seasonal cycle characterized by specific mixing and stratification conditions:

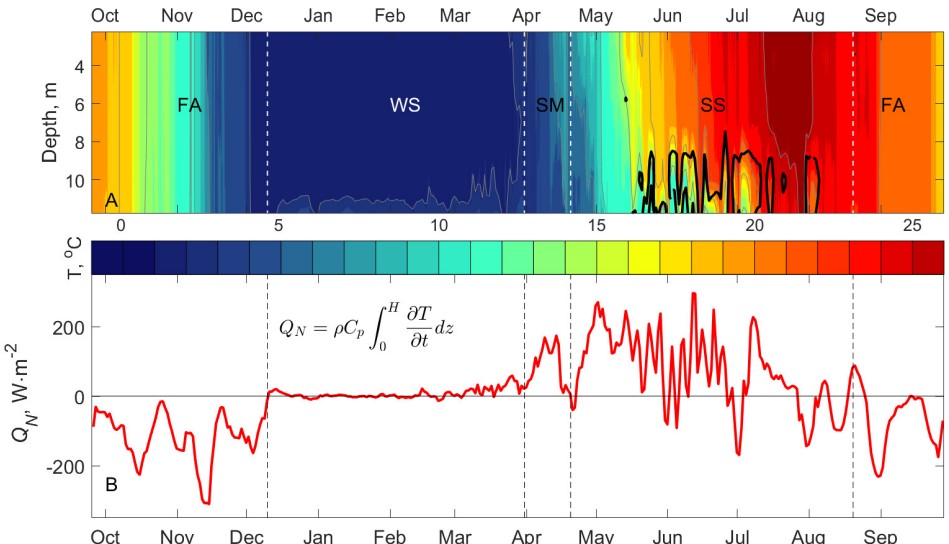

**Figure 3.** (A) Annual temperature cycle in the North Aral Sea. Thick black isoline is the boundary of the bottom stratified layer defined as $N^2 > 2 \times 10^{-3}\,\mathrm{s}^{-2}$. White vertical dashed lines mark the transitions of "lake seasons": FA refers to *fall (autumn) overturn*, WS to *winter stratification*, SM to *spring mixing* and SS to *summer stratification*. (B) Rate of change of the lake heat content (Net heat flux into the lake $Q_N$)

*Autumn-winter cooling (the fall/autumn overturn, FA in Fig. 3a)*, when convection by surface cooling effectively mixes the entire water column. The period starts in early September and lasts around 3 months until the ice cover formation. In our observations, the lake water column stayed thermally homogeneous from the start of the observational period on 21 September 2018 to the date of ice-on 09 December 2018 and from 01 September 2019 to the end of the observational period, and was characterized by a uniform decrease of temperature over time across the entire water column from about $18\,°C$ to $-0.44\,°C$ (which corresponds to the freezing point at $\approx 8\,\text{‰}$ for seawater salt composition). During this cooling phase, the daily mean surface-bottom temperature difference $\Delta T$ had a median value of $0.00$ $°C$ and did not exceed $0.2$ $°C$ in 80 % of days, interrupted by 1 to 2 daylong events of stratification with $\Delta T \leq 1\,°C$.

*Ice-covered period and winter stratification (WS in Fig. 3a).* The ice-covered period lasted from early December 2018 to 31 March 2019. First ice formed in shallow areas of the lake on 25-27 November 2018. On 06 December 2018 the ice cover started to form in the area of the autonomous measurement station that was reflected in a fast water temperature drop from





$\sim 0.7\,°C$ to $-0.44\,°C$ within 3 hours over the entire water column. On 11 December 2018, a stable ice cover formed, followed
by immediate increase of the near-bottom water temperatures and development of winter "inverse" stratification, characterized
by the downward temperature increase. The stratification had a two-layer structure with a thermally homogeneous mixed layer
at temperatures about -0.35 °C occupying the upper 10 meters of the water column, and $\sim 1.4\,\mathrm{K\,m^{-1}}$ downward temperature
increase down to the water depth of 12 m, where values of about $1.8\,°C$ were observed, which is about the maximum density
temperature $T_{md}$ as determined from the full mixing event in spring (see below). The lower stratified part of the water column
revealed quasi-periodic temperature oscillations with periods of 3 to 9 days.

The temperature and stratification regime changed remarkably in early March, when the snow cover melted and solar ra-
diation started to warm up the water under the ice cover and initiated convection in the upper water column by warmer and
heavier water sinking and mixing with the colder waters beneath, a phenomenon known from temperate and polar ice-covered
lakes (Mironov et al., 2002), and often referred as the "Winter II" season (Kirillin et al., 2012). From that moment the temper-
ature of the upper water column continuously increased with time from $< -0.3\,°C$ to $2.0\,°C$ within less than a month. At the
moment of ice breakup on 31 March, the heat gain from solar heating (estimated from the mean increase of the heat content
in the water column $Q_N = C_p \rho H \mathrm{d}T_{mean}/\mathrm{d}t$) amounted to $80\,\mathrm{W\,m^{-2}}$ (Fig. 3B). It is worth noting that temperatures at the
uppermost measurement depth of 2.2 m under the surface increased faster than below and created a vertical thermal instability.
The instability was not eliminated by convective mixing (cf. late March data in Figs. 3 and 4), which would be the case in a
thermally stratified freshwater lake, and persisted until the ice breakup. This fact can be interpreted as indirect evidence of the
water salinity contribution to the vertical stratification under ice: the apparent source of salt stratification is freshening of the
upper waters by the melt at the ice-water interface.

*Spring mixing (SM in Fig. 3a)* lasted in our observations from the ice breakup on 31 March to mid-April 2019. During this
period, solar and atmospheric heating at the surface causes the increase of surface temperatures from values close to freezing
point ($\sim -0.4\,°C$) to the temperature of maximum density and results in breaking of winter temperature inversion in the bottom
layer. It is worth noting that at the moment of ice breakup the mean water column temperature ($\sim 1.5\,°C$) is significantly higher
than the freezing temperature because of the solar heating under ice in previous months. Melting of the remaining ice floes
and heat release to the atmosphere produce short-term cooling and homogenization of the entire water column to $\sim 0.5\,°C$ on
01 April. Vertical thermal stratification with downward temperature decrease starts to develop around 09 April, when average
water temperatures arrive at $\sim 1.8\,°C$. Accordingly, this temperature represents the temperature of maximum density for the
salt composition and concentration of the North Aral waters.

*Summer stratification (SS in Fig. 3a)* lasted from late April to the end of August. After the initial nearly linear vertical thermal
stratification formed in late spring, the surface temperatures reached their maximum in late July with daily means of $> 25\,°C$
and peak values of up to 27 °C. Temperatures in the bottom layer remained generally lower with temperature differences
across the water column of 5 to $10\,°C$. However, the thermal stratification at the measurement site was repeatedly interrupted
by short-term events of full vertical mixing, indicating penetration of the surface waters down to the bottom (Fig. 5A). The
periodic character of these events suggest their wave origin. A deeper insight into the character of wave motions beneath the
mixed layer is provided by the spectral analysis of temperature oscillations combined with the theoretical model of basin-scale





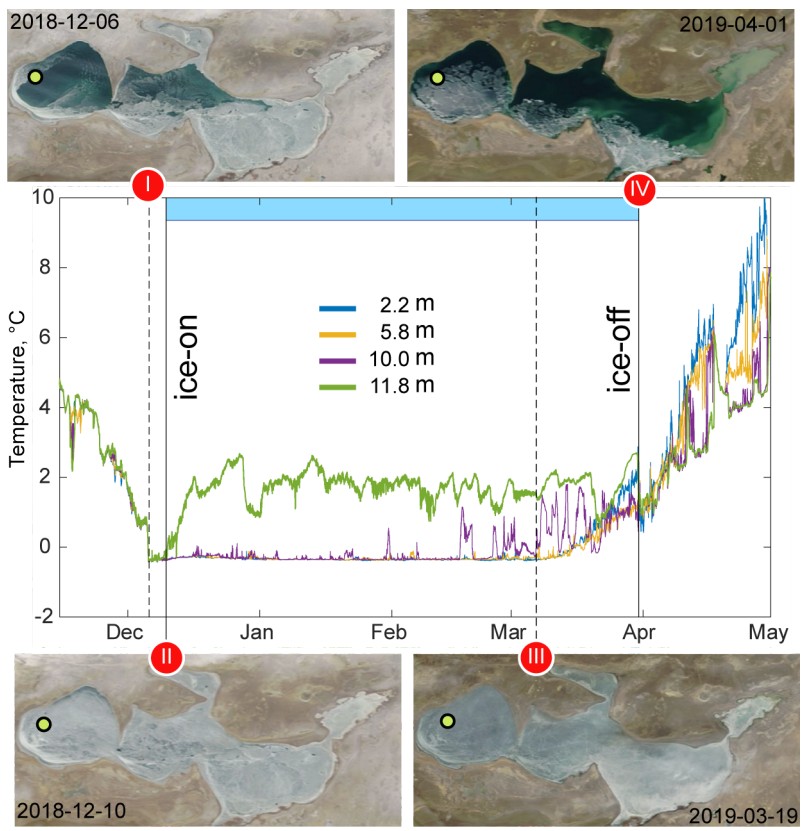

**Figure 4.** Temperature records between ice-on and ice-off in winter 2018-2019 at four selected water depths. The blue bar on top of the panel marks the ice cover duration. The four MODIS satellite images (NASA Worldview) with corresponding vertical lines across the temperature panel illustrate the ice conditions at the times of lake surface freezing [I-II], snow melt [III] and ice thaw [IV].

waves in a circular basin (Eqs. 5-6): the oscillations have periods of 4.5 days, which are slower than the inertial period $2\pi f^{-1}$
(Fig. 5B) and are therefore strongly affected by the earth rotation. According to the theoretical model, the period of 4.5 days in a basin with $L = 16$ km (characteristic radius of Shevchenko Bay) refers to the first-mode Kelvin wave with corresponding Burger number $S \approx 0.12$ (Eq. 4), the Rossby radius $S \times L \approx 2$ km, and wave speed $C_1 = 0.16\,\mathrm{m\,s}^{-1}$ (2). The same value of $C$ follows from the two-layered vertical temperature distribution (Eq. 3) in a 12 m deep basin with a 10 m thick upper mixed layer and water temperatures of 24 °C and 12 °C in the upper and the lower layer, respectively. These values agree well with
the vertical temperature profiles observed at the monitoring site (Fig. 5A).

For a Kelvin wave with frequencies $\sigma << f$ the theory predicts a counterpart Poincaré wave to exist at frequencies $\sigma \geq f$. A statistically significant peak at the second frequency from Eqs. (5)-(6) corresponding to a Poincaré wave exists in the temperature spectrum, although it is much less energetic than that for the Kelvin wave. On the one hand, this result is evidence that the observed temperature oscillations were produced by a Kelvin-Poincaré pair of basin-scale waves. On the other hand,
it demonstrates that the contribution of the Poincaré wave to the oscillations was weak. The latter conclusion is confirmed by



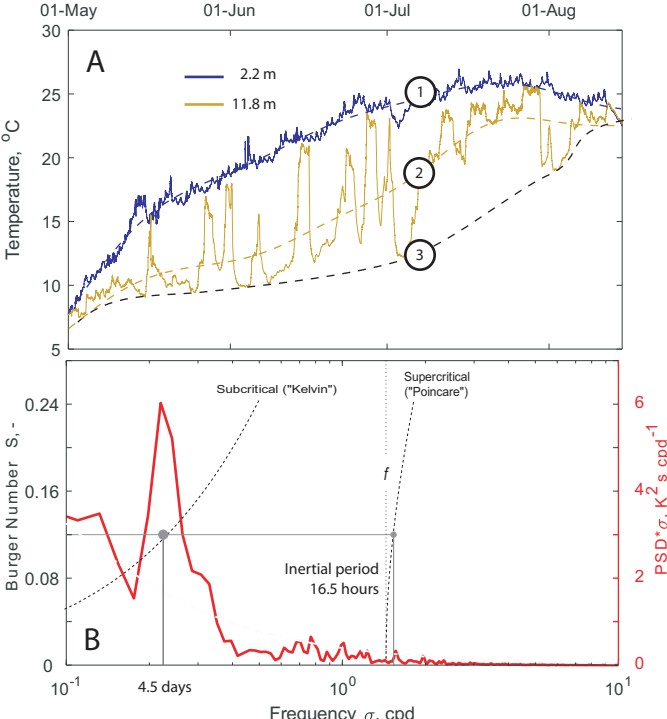

**Figure 5.** Summer temperature stratification and internal wave characteristics. (A) Time series of the [1] surface and [2] near-bottom temperatures in summer 2019. The bottom temperatures are complemented by the lowest temperature envelope [3] as a proxy of temperature of cold deep water masses ascending to the measurement site. (B) Spectral density of near-bottom (11.8 m depth) temperature oscillations. The dashed line corresponds to the lower confidence boundary for the spectral peaks defined as the upper 95 % for the red noise signal (first order autoregressive process) containing the same variance as the observed time series (Gilman et al., 1963). Black dotted lines show the Burger number dependence of the wave frequency $S(\sigma)$ following Eqs. (5)-(6), and the inertial frequency $f$.

previous studies (Antenucci and Imberger, 2001; Kirillin et al., 2009): while Kelvin waves have their their maximum energy at the lateral boundaries, with amplitudes decreasing exponentially towards the lake center, the amplitudes of Poincaré waves are concentrated in the horizontal-plane circular motions around the mid-part of the lake and weakly affect the vertical motions of isotherms.

A closer insight into external forcing behind the temperature conditions in the North Aral is provided by the rate of change of the bulk heat content in the lake $Q_N = \rho_0 C_p \int_0^H \partial T / \partial t \, \mathrm{d}z$, where $C_p$ is the water heat capacity. The value represents a measure of the net heat flux to the water column (Fig. 3B) and demonstrates strong seasonal flux variations reflected in the thermal stratification. During the autumn cooling the lake lost energy at $\sim 100 \ \mathrm{W \, m^{-2}}$ (0.3 K/d lake-mean cooling rate), with peaks as low as $-300 \ \mathrm{W \, m^{-2}}$. The average heating rates in spring amounted to $\sim 100 \ \mathrm{W \, m^{-2}}$, while summer fluxes

experienced strong variations with occasional negative values, which can be ascribed to cold water advection by basin-scale internal waves rather than to surface cooling events.



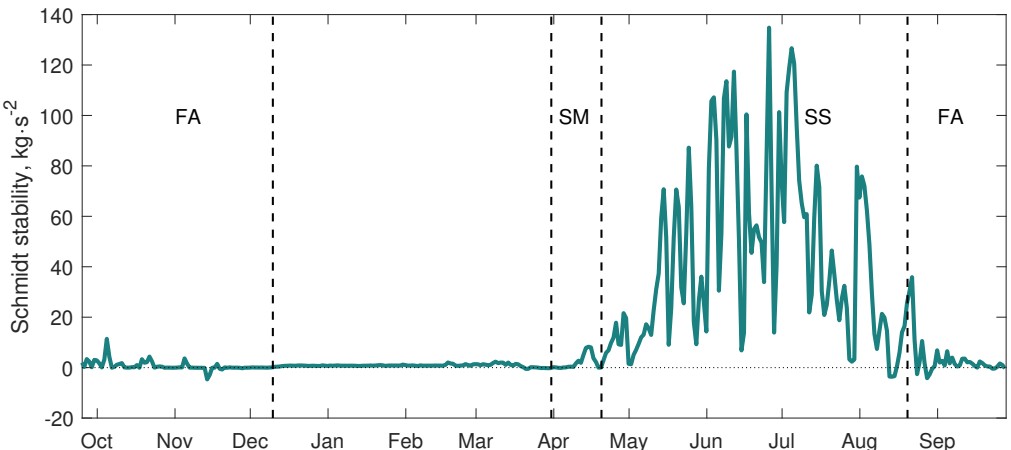

**Figure 6.** Annual course of thermal stratification in the North Aral Sea expressed in terms of Schmidt Stability (Eq. 7)

Apart from the summer stratification period, the water column remained well-mixed vertically with the Schmidt stability $St$ close to zero. In summer, $St$ followed the internal wave-driven temperature fluctuations at the measurement site (Fig. 6) varying between 0 and $> 100 \, \mathrm{kg\,s^{-2}}$. The maximum values peaked at $St \approx 130 \, \mathrm{kg\,s^{-2}}$, which is about 2 times lower than those typical for seasonally stratified dimictic lakes (Read et al., 2011). Hence, a typical surface wind stress $\tau \approx 0.05 \, \mathrm{N\,m^{-2}}$, roughly corresponding to wind speeds of 4 to 6 $\mathrm{m\,s^{-1}}$, would result in the isotherms slope $\tau h_{eq} St^{-1} \approx 1.5 \, \mathrm{m\,km^{-1}}$. Approximating the Shevchenko Bay by a cylindrical shape with radius $L \approx 16$ km and the height of the surface mixed layer $h_{mix} \approx 10$ m, the ratio $(\tau h_{eq}/St)(h_{mix}/L)^{-1} \approx 2 > 1$, indicating that the slope would produce surfacing of the deep waters at the upwind shore and could potentially destroy vertical stratification completely. The persistence of stratification throughout the summer can be attributed to the irregular bathymetry of the North Aral Sea and the nonlinear slope of the isotherms influenced by the Coriolis effect, which transforms the wind energy to rotational wave motions with the isotherms slope concentrated within a distance of the Rossby radius $L_R \approx 2$ km from the lateral boundaries (Lamb, 1932; Gill, 1982; Antenucci and Imberger, 2001).

### 3.3 Observed oxygen regime

Our year long DO record provides first insights into seasonal DO dynamics in the North Aral Sea and demonstrates a generally high saturation of the lake waters with oxygen in the annual cycle (Fig. 7): during most of the year, the DO at 12 m water depth remained at around 100 % of saturation, whereas the mean absolute concentrations varied between ∼15 ppm in winter and ∼8 ppm in summer reflecting the seasonal variations in water temperature and the corresponding saturation levels. An increase of DO concentration from 10 ppm to 15 ppm was observed during autumn-winter cooling caused apparently by increase of solubility due to water cooling and absorption of oxygen from the atmosphere facilitated by strong convective mixing (cf. the typical cooling rates of ∼100-200 $\mathrm{W\,m^{-2}}$ in October-November in Fig. 3). The DO saturation during the ice-covered period was slightly above 100 % and increased with time, suggesting the existence of an internal DO source in the lake despite its isolation from the atmosphere.





From early March, coinciding with the start of under-ice heating by solar radiation ("Winter II" period), the DO concentrations decreased from 15 ppm to the local minimum of 13 ppm (from 105 % to 90 % of saturation) around 15 March. The
decrease was accompanied by oscillations with amplitudes of 2 to 4 ppm and periods of about 3 days. Parallel oscillations were present in temperature records at the same water depth of $\sim 10$ m (cf. Period III in Fig. 4). Later, the mean concentrations stabilized at $\sim 14$ ppm, while the DO saturation gradually increased to 120 % in mid-May. In turn, the long-period (>1 day) oscillations persisted after the ice break, with a short interruption during the spring mixing on 15-20 April, and resuming in summer with a lower intensity.

This behavior demonstrates the close connection between oxygen dynamics and stratification conditions. The observed DO fluctuations are directly linked to temperature oscillations caused by internal basin-scale waves and can be ascribed to wave-driven events of colder undersaturated water upwelling from deeper lake areas to the measurement location. The long period summer DO oscillations have the same periods of 4.5 days as the temperature oscillations associated with the Kelvin wave (cf. low frequency spectral peaks in Figs. 8 and 5). The similar oscillations at the end of the ice-covered period had apparently
the same wave-driven character. As demonstrated by the temperature data, after the snowmelt, solar radiation supplied a significant amount of energy to the ice-covered water column. A part of this energy was spent on heating the upper waters, while another part was transformed into kinetic energy of vertical convective motions and spent on melting the ice at its base. Thus, simultaneous heating, vertical mixing, and freshening of the upper water column created a complex vertical density stratification as a background for development of basin-scale internal waves under ice (Kirillin et al., 2009). These waves with
periods essentially longer than the inertial period should have the same rotational character (see Eqs. 5-6) as those observed in summer (cf. Fig. 5). Interruption of oscillations in the spring mixing period, when stratification was destroyed, also points to internal waves as the origin of the oscillations. In general, the DO dynamics indicate the presence of a water mass with lower oxygen concentrations in the deep central part of the lake during both summer and winter stratification periods. Driven by basin-scale internal waves, these deep waters periodically flowed over the measurement site and caused DO saturation drops
down to about 60 %.

Spectra of near-bottom DO variations in summer (Fig. 8) reveal a significant peak at 24 hours period. An apparent origin of diurnal variations in DO content is the primary production (PP) in the water column and(or) at the lake bottom. The up to 120 % oversaturation in May-June can be also attributed to PP during the spring phytoplankton bloom as its most probable source. The appreciable effect of PP on DO content at 12 m water depth is a remarkable feature indicating sufficient availability
of light and nutrients over the major part of the North Aral bottom. The diurnal oscillations were however absent in late winter DO fluctuations (not shown), suggesting limited light availability at the 12 m water depth during this period.

### 3.4 Modeled thermal regime

The results of the uncalibrated model FLake validation showed an excellent fit to the observed surface temperature over the whole measurement period ($RMSE = 0.80$ °C). The prediction of the ice covered period was equally good. The model
predicted stable ice duration from 13 December 2018 to 3 April 2019, thus overestimating ice-on and ice-off dates by only 2-3 days, and the total ice duration (110 days) by only 1 day. The modeled bottom temperature closely followed the observed





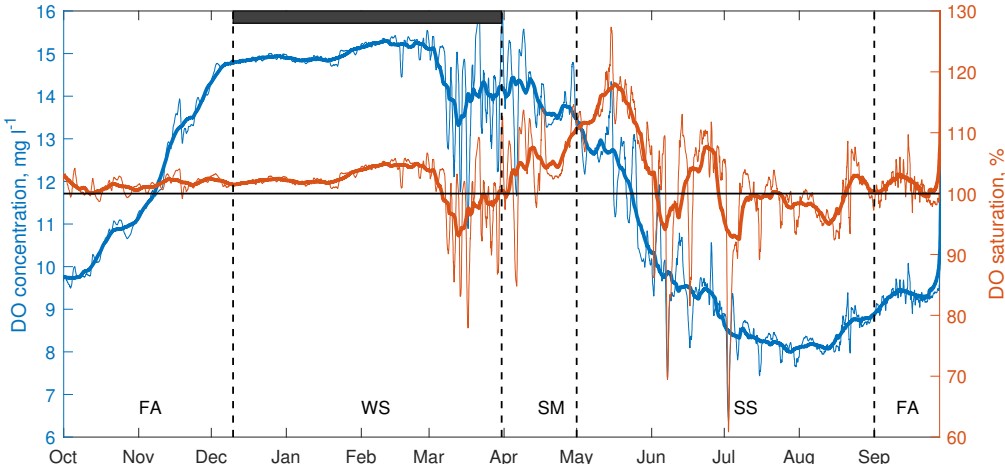

**Figure 7.** Annual cycle of dissolved oxygen at 12 m water depth. Values of DO concentration in ppm are shown with blue line, while values of DO saturation are shown with brown line. Black vertical dashed lines mark the transitions of "lake seasons": FA refers to *fall/autumn overturn*, WS to *winter stratification*, SM to *spring mixing* and SS to *summer stratification*.

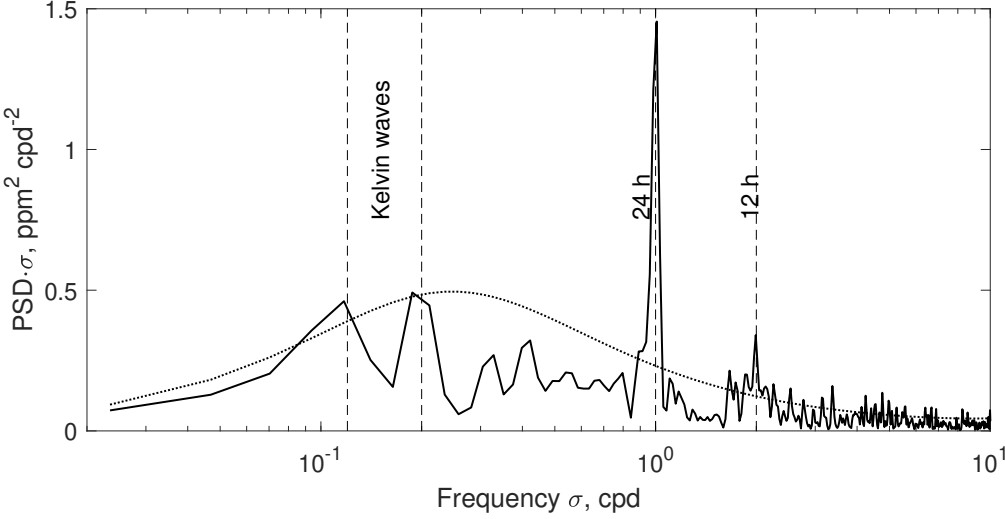

**Figure 8.** Spectral density of near-bottom DO content oscillation in summer (June-Aug). Vertical dashed lines mark the periods of significant energy-containing oscillations. The dotted line shows the lower confidence boundary for the spectral peaks defined as the upper 95 % for the red noise signal (first order autoregressive process) containing the same variance as the observed time series (Gilman et al., 1963).





bottom temperature ($RMSE = 2.60\,^{\circ}\mathrm{C}$) when the mean lake depth was set to 7 m (Fig. 9a), corresponding to the mean depth of the whole North Aral Sea according to satellite altimetry (about 7.3 m). However, the one-dimensional model expectedly failed to reproduce the observed bottom temperature fluctuations with periods of several days, produced by wave-driven advection

of water along the bottom slope. As a result, the model predicted a dimictic regime of the bay with stable, continuous summer stratification and vertical temperature differences from 5 to 17 $^{\circ}\mathrm{C}$, in contrast to the intermittently interrupted stratification that was observed. On the other hand, if the water level was decreased by 1 m to a mean mean lake depth of 6 m, the model predicted polymictic behavior with the bottom temperature equal to the surface temperature throughout the year. The result suggests a high sensitivity of the mixing regime in North Aral to water level fluctuations. Sensitivity model runs with a stepwise increase

of the lake depth up to 14 m (Fig. 9a) revealed a gradual decrease of the mean bottom temperature in summer from $\sim$15 $^{\circ}\mathrm{C}$ to $\sim$5 $^{\circ}\mathrm{C}$ and a slight increase of the summer stratification duration. We performed a sensitivity analysis to explore the combined effect of changes in water depth and another length scale, the light extinction (Secchi depth) on the thermal regime, where the variable lake depth simulates potential variations in the lake level due to water regulations at the lake catchment and at the dam, and extinction variations reflect the bulk effects of changes in trophic level (productivity) on vertical redistribution of solar

radiation over the water column. The thermal regime appears to be very sensitive to both a change in water column depth and water transparency (Fig. 9b), with both polymictic and dimictic regimes possible. The model suggested that North Aral would be polymictic at lower depth and light extinction, and dimictic at higher depth and extinction. Further, the model suggested that North Aral would switch from a polymictic to a dimictic regime at a lake depth around 6 to 9 m and corresponding extinction between 0.3 and 0.8 $\mathrm{m}^{-1}$. With the mean depth of 7 m and extinction of 0.5 $\mathrm{m}^{-1}$, North Aral is currently close to a transitional

state in the mixing regime. The result also suggests that changes in the regional water budget or modifications of the Kokaral Dam that affect water level, or changes in water quality that affect water transparency, may considerably affect the thermal regime. For instance, a 1 m increase in depth from 7 to 8 m would increase stratification duration by 40 days and the July vertical temperature difference by about 3 $^{\circ}\mathrm{C}$ according to the model. A depth increase from 10 to 11 m on the other hand would increase stratification by 15 days and the July temperature difference by 2 $^{\circ}\mathrm{C}$.

## 4 Discussion


Our results demonstrate significant changes in the physical and chemical conditions in the North Aral Sea after construction of the Kokaral Dam. Salinity of the North Aral waters has dropped significantly, by almost 7 $\mathrm{g\,kg}^{-1}$ and is currently stabilized around $\leq 11\,\mathrm{g\,kg}^{-1}$, which is close to the brackish conditions observed during the "natural" period before the shrinkage (Bortnik and Chistyaeva, 1990; Kosarev and Kostianoy, 2010). Apart from the manifold consequences for the biotic components of

the lake ecosystem, which are out of the scope of our study, the salinity decrease implies several important consequences for the seasonal mixing and oxygen regime of the lake. These consequences are manifested primarily through the salinity effects on the freezing temperature $T_f$ and the temperature of the maximum density $T_{md}$. Our data on vertical mixing allowed these values to be estimated in the North Aral Sea as $T_f \approx -0.4\,^{\circ}\mathrm{C}$ and $T_{md} \approx 1.8\,^{\circ}\mathrm{C}$. These estimates correspond to the observed salinity of 10.5 $\mathrm{g\,kg}^{-1}$ and slightly differ from the average values of -0.57 $^{\circ}\mathrm{C}$ and 1.60 $^{\circ}\mathrm{C}$ respectively, given for the entire



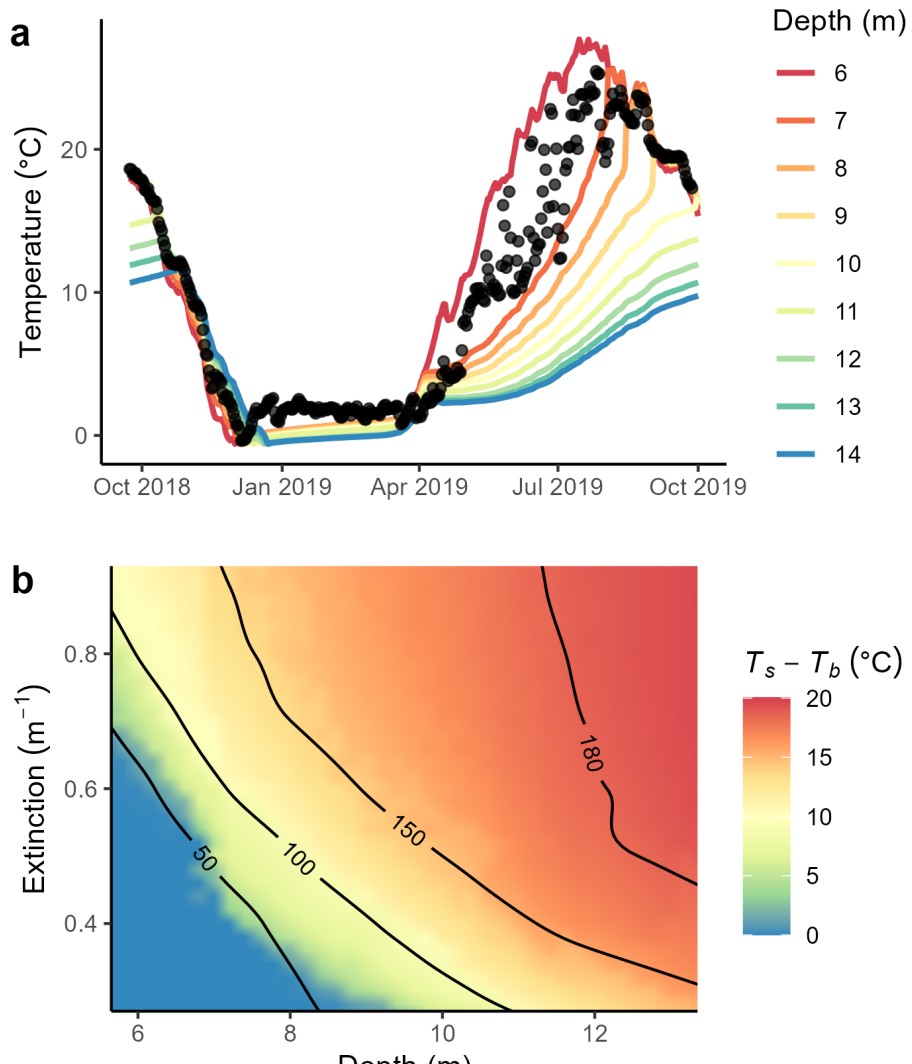

**Figure 9.** FLake model results and sensitivity of the thermal regime to lake depth and light extinction. (a) Observed bottom temperature at 11.8 m (black dots), and modelled bottom temperatures at lake depths ranging between 6 and 14 m (coloured lines). (b) Modelled mean temperature difference between surface ($T_s$) and bottom ($T_b$) in July as a function of mean lake depth and light extinction. The black contour lines indicate the corresponding summer stratification duration in days.



lake at the salinity of $10\ \mathrm{g\,kg^{-1}}$ during pre-desiccation conditions (Bortnik and Chistyaeva, 1990). The observed difference can also be linked to a transformation of the salt composition, which occurred during the lake shrinkage and separation of the Aral's residual basins from each other.

In contrast to ocean waters and hypersaline lakes, brackish lakes have a temperature of maximum density above the freezing temperature, $T_{md} > T_f$, which causes strong convective mixing during the autumn cooling period and homogenization of
the entire water column. As a result, the effect of salinity on the vertical density stratification is minor compared with that of temperature, which prevents formation of permanent deep water stagnation (meromixis) as observed in the southern parts of the Aral Sea (Izhitskiy et al., 2021). This fact also implies presence of warmer waters near the bottom during winter stratification, which is expected to be common in large brackish lakes of arid climate, in contrast to the nearby hypersaline waters of the Southern Aral Sea. Hence, like in the past natural state, autumn overturn is one of the most important processes forming the
present hydrological structure of the lake.

On the one hand, relatively low salinity ensures formation of the seasonal ice cover in winter, which is completely absent in the hypersaline southern Aral Sea (Kouraev et al., 2004; Kouraev and Crétaux, 2010). On the other hand, brackish waters freeze at lower temperatures than freshwater: this is one of the factors explaining low bulk water temperatures during the ice-covered period. Another factor is a strong heat loss during the autumn cooling before the ice formation, typical for the continental
(cold desert) climate of Central Asia. In this regard, the autumn and winter regime of the North Aral Sea is similar to that of freshwater lakes of the Tibetan Plateau, where intense surface cooling in autumn is accompanied by strong winds preventing ice cover formation (Kirillin et al., 2021).

Despite the weak vertical salinity gradients in the the brackish North Aral Sea, these gradients may still affect vertical mixing at certain stages of the seasonal stratification. In contrast to freshwater lakes, convective mixing during the autumn-
winter cooling occurs in two stages: the first stage begins with the onset of surface cooling, characterized by strong convective mixing, which continues until the mixed water layer reaches the temperature of maximum density. Afterwards, convection due to the surface heat release ceases. In freshwater lakes, the inverse stratification starts to form at this moment (Kirillin et al., 2021). However, after the water temperatures reaches the freezing point and the ice cover starts to form, the second phase of convective mixing begins, driven by the density increase near the surface due to salt exclusion by ice formation (Pieters and
Lawrence, 2009). This effect is more pronounced in brackish waters and may significantly contribute to vertical mixing in early winter (cf. Stage I in Fig. 4). Vertical salinity gradients have an opposite effect in late winter with the onset of under-ice heating by solar radiation. In freshwater lakes, the resulting increase of the surface water temperatures produces convective mixing (Matthews and Heaney, 1987; Mironov et al., 2002), which is prevented in the brackish conditions by freshening of surface waters due to melting at the ice base, similarly to the oceanic ice boundary layer (McPhee, 1992). It is worth noting that the
period of under-ice heating in North Aral lasts for almost 2 months, which is significantly longer than in temperate and polar lakes (Mironov et al., 2002; Bouffard et al., 2019). The long duration of this phase of the ice-covered period (sometimes termed "Winter II") is a distinguishing feature of low-latitude lakes, which gain a larger amount of radiation than typical temperate and (sub-)arctic ice-covered lakes. In this regard, North Aral shares the features of the lakes of Tibet and the Mongolian Plateau, which are characterized by strong heating of the under-ice water column (Kirillin et al., 2021; Huang et al., 2022). Accordingly,



the late winter is a crucial period in these lakes, with a warm and quiet environment under ice favorable for planktonic primary production and fish.

The cold arid climate is characterized by the strong seasonal variations in the heat flux at the lake-atmosphere interface resulting in a wide seasonal range of lake temperatures. Before the desiccation, the range of seasonal temperature fluctuations at the Aral surface did not exceed 23-25 °C (Bortnik and Chistyaeva, 1990), whereas now this range exceeds 27 °C. This is

apparently due to the significant decrease of the lake volume, and, as a result, the lower heat storage capacity of the reservoir. This fact also explains the more intense cooling and heating processes observed during the autumn and spring mixing period: the net rates of surface cooling in autumn and surface warming in summer exceed $200 \, \mathrm{W\,m^{-2}}$, which is several times higher than the values reported in temperate lakes (Henderson-Sellers, 1986).

The relative shallowness of the North Aral Sea determines another characteristic feature of its seasonal mixing regime:

despite the strong surface heating, the summer stratification is relatively weak, as demonstrated by the low Schmidt stability. The latter suggests that moderate winds could completely destroy the summer stratification, turning the lake to *polymictic*, i.e. fully mixed vertically during most of the year (Hutchinson, 1957). However, our analysis of seasonal temperature time series in time and frequency domains reveals a 2-3 m thin near-bottom layer with temperatures decreasing downwards, which is persistent throughout the summer period. Due to the decrease of the surface area with depth, the relative volume of the layer is

small compared to the volume of the entire lake. As a result, the wind-induced basin-scale internal waves, which are inevitably produced in the stratified water column, move the entire cold near-bottom water along the bottom slope instead of mixing it with the overlying warm mixed layer (see mid-panel in Fig. 10). Thanks to the large horizontal dimensions of the lake and the low water column stability, the period of these waves is rather long, reaching several days, which significantly exceeds the inertial period at the lake latitude. Hence, the Coriolis effect captures the wind energy in the rotational motion instead of vertical mixing

(Lamb, 1932; Gill, 1982). Thereby, the large horizontal dimensions and the weak thermal stratification prevent the lake from becoming polymictic. Apart from preventing full vertical mixing, the basin-scale waves play a potentially important role in the water-sediment mass exchange by enhancing shear mixing in the bottom boundary layer and increasing the supply of dissolved oxygen to the sediment surface and transport of dissolved nutrients from the sediment to the water column. This suggestion is supported by our data on near-bottom DO concentrations: the waters at 12 m depth remain well-saturated with oxygen in

summer, while short-term wave-driven drops of concentrations to ∼60 % of saturation indicate existence of a low-oxygen volume in the deepest part of the lake, which however is small relative to the entire volume of the lake. The specific thermal and mixing conditions determine also the oxygen regime in winter: the DO saturation under ice is about 100 % throughout the ice-covered period, indicating no current risk of strong hypoxia in North Aral, distinguishing it from the majority of temperate and boreal lakes, where winter hypoxia is a common phenomenon with negative consequences for lake ecosystems (Malm,

1998; Leppi et al., 2016; Steinsberger et al., 2020; Perga et al., 2023; Schwefel et al., 2023). A distinct feature of the under-ice oxygen regime in the North Aral Sea is the high absolute DO concentrations, reaching ≥ 15 ppt. These values are 15-25 % higher than typical concentrations in freshwater lakes at higher latitudes (Terzhevik et al., 2010; Bengtsson, 2011; Bouffard et al., 2013) and are due to the low under-ice temperatures of the brackish lake water and the corresponding high DO solubility.




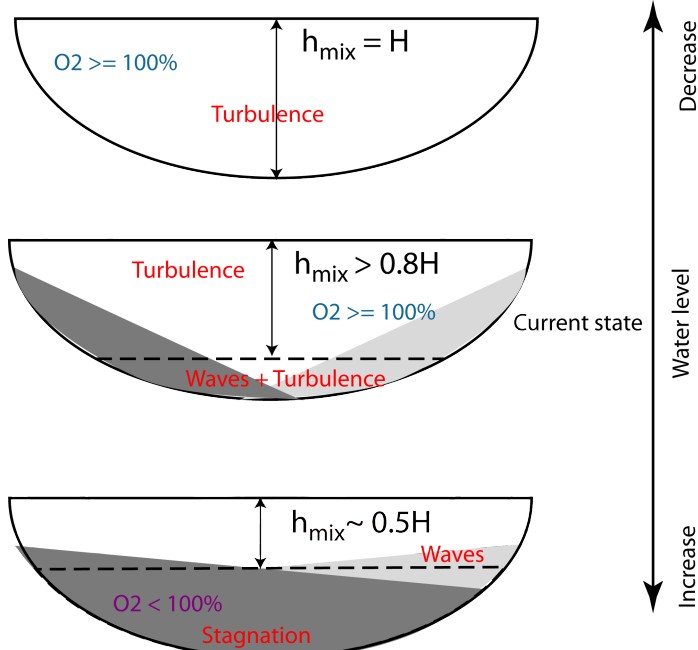

**Figure 10.** Sketch illustrating the current state of the vertical mixing regime in the North Aral Sea and possible scenarios of its transformation depending on the water level fluctuations. $h_{mix}$ is the thickness of the upper mixed layer, $H$ is the lake depth, $O2$ is the dissolved oxygen saturation level.

Therefore, brackish seasonally ice-covered lakes, which are common for the endorheic zone of Eurasia, are more effective in
storing oxygen under ice and are less prone to the oxygen deficit in winter than temperate freshwater lakes.

Summarizing our results, we can conclude that the North Aral Sea is now in a transitional mixing regime between dimictic and polymictic. This state is intrinsically unstable and is prone to a quick regime shift. As demonstrated by the model experiments, such a shift can be triggered by variations of the water level or by a change of the water transparency due to e.g. increase of primary production. The sensitivity model runs showed that under the currently observed stratification, a change in depth
of 2 m can shift the mixing regime and potentially alter stratification duration by up to two months. This magnitude of water level variations is not unrealistic: According to recent assessments of the variability of water balance conditions in the North Aral Sea, the discharge of the main inflow, the Syr Daria River, will undergo significant fluctuations in the future (Ayzel and Izhitskiy, 2018, 2019). Combined with the variability of evaporation and precipitation, this could lead to significant changes in the lake surface level. According to climate scenarios, the lake surface may drop by several meters by the end of the century
(Izhitskiy and Ayzel, 2023). In this case, a change of the seasonal mixing regime to polymictic (Fig. 10) can be expected. While the risk of deep hypoxia would be minimal in this case, negative consequences for the ecosystem may emerge as a strong increase of primary production and eutrophication of the lake due to increase of the nutrient concentration. On the other hand, the water authorities are considering further resortation measures including raising the Kokaral Dam crest by another 6 meters





(Izhitskiy and Ayzel, 2023). The resulting water level increase would enhance the vertical thermal stratification, eventually
turning the seasonal mixing regime to dimictic with the summer stagnation period lasting for several months (Fig. 10). This
scenario suggests development of deep hypoxia, or even anoxia with long-lasting oxygen-free conditions near the lake bottom.
These possible changes in mixing regime can have severe consequences for the biogeochemical cycle of the lake. Currently,
concentrations of dissolved methane in North Aral exceed atmospheric equilibrium by a factor of 13 in the southern part of
the lake, and by a factor of 48 in the area near the Kokaral Dam (Izhitskaya et al., 2019). High inflows of riverine organic
matter (Klimaszyk et al., 2022) and its reduction in anaerobic microenvironments and sediments in combination with a possi-
ble increase of stratification may lead to a strong increase of anoxic methane production near the lake bottom, similar to that
observed currently in the Western South Aral Sea and Lake Chernyshev (Izhitskaya et al., 2019). Our measurements of Chl-a
concentrations suggest the North Aral Sea to be currently in a mesotrophic state. The mesotrophic character of the lake is also
supported by our Secchi disk measurements of 3.5 m and by the trophic index estimates based on inorganic phosphorus and
nitrogen by Klimaszyk et al. (2022). These authors also found a strong increase of the nutrient concentrations towards the river
inlet, which indicates a high nutrient supply by the inflow. Therefore, a possible increase of water level and subsequent transi-
tion to dimictic conditions can strongly increase nutrients retention in the lake and provoke transition to eutrophic conditions
(Huisman et al., 1999), which can be accelerated by the plankton feedback on the mixing conditions (Shatwell et al., 2016).

## 5 Conclusions

Terminal lakes are sensitive indicators of anthropogenic activity, such as water use for irrigation, and of the climatically driven
changes in the hydrological regime of endorheic areas. The area of the Aral Sea catchment is $3 \times 10^6 \, \text{km}^2$ (Zavialov, 2007),
which makes it the second largest endorheic catchment after the Caspian Sea. In this regard, the results of the Aral Sea
conservation experiment represent a unique opportunity to assess the outcomes of large-scale lake manipulation for its thermal
and mixing regime. The evident decrease of salinity and restoration of the biological communities including the fish population
after the dam construction were acknowledged as indicators of the success of the conservation experiment. While the observed
temperature and salinity regime of the North Aral Sea is now similar to the pre-desiccation state of the lake, our results
demonstrated distinct mixing characteristics on seasonal and sub-seasonal time scales, which make the regime unstable. This
unstable mixing regime favors on the one hand heat and mass exchange at the lake bottom, but can be quickly disturbed by
further discharge manipulation or by consequences of global change.

*Data availability.* The data is temporarily available for reviewing purposes under https://nimbus.igb-berlin.de/index.php/s/SA5NT7cMQctNgtf.
After the study is published, the dataset will be assigned a DOI and made freely available via the Freshwater Research and Environmental
Database (FRED) of the Leibniz-Institute of Freshwater Ecology and Inland Fisheries, Berlin.



*Author contributions.* AI and GK conceived the study; GK, TS, and AI designed and performed field observations; TS designed and performed model experiments; AI and GK analyzed the field data; AI wrote the first version of the manuscript; GK wrote the final version with contributions from all co-authors.

*Competing interests.* the authors declare no competing interests

*Acknowledgements.* The study is part of the research project "LaMer" funded by the German Research Foundation (DFG, project id KI 853/16-1). The fieldwork was supported by the state agreement of the Shirshov Institute of Oceanology RAS No FMWE-2024-0015. TS was partially supported by the German Research Foundation (DFG grant no. SH 915/1-1). Additional support by the DFG PRoject "PycnoTrap"

(DFG grant no. GR 1540/37-1) is thankfully acknowledged by GK. We thank all participants of the joint fieldwork on the Aral Sea 2016-2019. We also acknowledge the use of imagery provided by services from NASA's Global Imagery Browse Services (GIBS), part of NASA's Earth Observing System Data and Information System (EOSDIS).





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
