# Peer review of "Consequences of the Aral Sea restoration for its present physical state: temperature, mixing, and oxygen regime"

_EGUsphere, 2025_

## Referee Comment (RC1)

**Review of Kirillin et al. (2025)** - **Consequences of the Aral Sea restoration for its present physical state: temperature, mixing, and oxygen regime**

This study investigates the current mixing dynamics of the western basin of the North Aral Sea (Shevchenko Bay), 15 years after a dam was built as an attempt to save the North Aral Sea from desiccation. The study demonstrates from year-round measurements of thermal stratification and oxygen that a thin and low oxygen bottom stratified layer persists in summer as in dimictic lakes. However, inertial internal waves move the stratified layer around, causing the sloping sides of the Shevchenko Bay to be fully mixed several times a year as in polymictic lakes. The authors call this mixing regime "transitional" between dimictic and polymictic and show with 1D numerical simulations that an increase (decrease) in water level and light attenuation would bring the lake to a steady dimictic (polymictic) regime.

Although the physical processes quantified by the study are well known, the fact that they are observed in a unique, evolving system makes the findings of the study very interesting and suitable for publication in HESS. The analysis that combines year-round temperature and oxygen measurements, CTD profiles, satellite images and 1D numerical modelling is comprehensive and robust. Yet, several statements are misleading or lack detailed explanation. Most of my comments aim at improving the clarity and readability of the manuscript.

**Specific comments :**

1. The location of the measurements is different from typical monitoring at the deepest point of the lake, which should be more emphasized through the manuscript. In fact, the data has been collected on the **sloping sides** of the **western basin** of the North Aral Sea (Shevchenko Bay; Fig. 1), which (i) might not be representative of the other basins of the North Aral Sea (including the deeper, central basin) and (ii) differs from measurements at the deepest point of the Shevchenko Bay. The authors should clearly state in the abstract, results, and discussion that their analysis focuses exclusively on the western lake basin. Additionally, they should elaborate on any potential differences or similarities in mixing dynamics between the western and central basins. Knowing the presence of an east-west salinity gradient (l. 241) and a ~7 m deep sill between the two basins (Fig. 1), do the authors expect wind-driven or density-driven basin exchange that could have affected their measurements? The mooring misses the bottom 25% of the water column because it was not deployed at the deepest point of the Shevchenko Bay. An additional mooring at the deepest point would have provided essential information on the full water column, including the persistence of the bottom stratified layer in summer. The authors should clearly indicate this limitation in the methods and that the frequent mixing events (polymictic-type) were only observed on the sloping sides of the basin.

2. It would help the reader to better explain the past, current and future mixing regimes of the Aral Sea. First, explicitly define "dimictic" and "polymictic" where the mixing regime is introduced in l. 61-67 of the introduction and remove the word "dimictic" in l. 43. The comparison between the natural (dimictic), pre-restoration (polymictic?) and current (transitional) mixing regimes of the Aral Sea should also be clearer by (i) providing information in the abstract about the past mixing regime (l. 18) underline before describing the current mixing regime in l. 10 and (ii) using the natural mixing regime as a reference point through the manuscript. The end of the abstract and the conclusion do not provide a clear answer regarding the preferred regime: is the goal to return to the natural, dimictic regime?
The current, "intermediate" (l. 10) or "transitional" (l. 409, 496) mixing regime should be better explained as it could be incorrectly interpreted as a regime alternating between polymictic and dimictic temporally (i.e., between years), whereas this variation is spatial

(i.e., frequent mixing on the sloping sides, persistent stratified layer at the deepest point). As explained in my first comment, the authors should provide hypotheses about the persistence of summer stratification at the deepest point. They could estimate the percentage of the Shevchenko Bay that remains stratified (based on a 2-3 m stratified layer, l. 474) instead of using the term "small" (l. 11, 475, 486).

Regarding the future mixing regime, I suggest to explicitly mention the effects of an increase (or decrease) in water level and light attenuation instead of referring to "changes" or "shifts" which remain vague (for example in l. 20 and l. 96).

3. The information about the morphology of the Ara Sea and its history should be re-organized in a more logical way. Volume and surface areas given in the abstract (l. 2-3) and in the conclusion (l. 526-527) should be moved to the introduction where the Aral Sea is presented (e.g., l. 50-54). A few characteristics of the restoration project (e.g., construction of a dam) should be given as soon as the "Aral Sea restoration project" is mentioned in l. 37. The name of the river inflows and the dam must be given as soon as they are introduced in l. 40 and l. 48, respectively. The paragraph about the climate of the Aral Sea region (l. 84-88) should be moved earlier in the introduction, maybe just after the mixing regime has been introduced (l. 67). The importance of the North Aral Sea (l. 88-92) should also be stated earlier, where it is introduced in l. 40 for example. Differences between the Southern (hypersaline) and Northern (brackish) Aral Sea could also be developed in the introduction so that the comparison between the mixing dynamics of the Northern and Southern Aral Sea (l. 428-435) would be clearer. Is $T_{md} > T_f$ in the Southern Aral Sea?

4. The assumption that salinity has negligible effects on stratification (l. 132-133) must be more justified and nuanced. Could the authors compare estimates of vertical bottom density gradient from conductivity profiles and from temperature profiles (e.g., June CTD profile) to support their assumption? Although salinity-driven stratification is negligible in summer, salinity effects might still play a role during turnover periods, when water temperature reaches the temperature of maximum density. It seems that an inverse thermal stratification occurred in mid-November (Figs. 3a, 4), which would indicate a role of salinity in maintaining a (weak) stratification. Salt exclusion could also generate convection (ice growth) or stratification (ice melt) in winter. The model does not include this process (l. 201-203), but l. 448-454 state that salt exclusion is important in brackish lakes, which seems inconsistent.

   In addition, the model assumes constant salinity over time (l. 203), which differs from the continuous decrease in salinity observed after the dam construction (Fig. 2) and from the natural seasonal variability of salinity (l. 76-77). The authors should mention this model limitation and discuss the potential effects of a further decrease in salinity on the mixing dynamics: would a decrease in salinity reduce the thermal expansivity of water (Eq. 10) and the tendency of the lake to stratify?

5. The methods section lacks the following information for better clarity:
   - Maximal and mean depth of the North Aral Sea and the Shevchenko Bay.
   - Value of the gravity acceleration in l. 127 (instead of l. 216).
   - Formula of $g'$ and $h_{eq}$ in l. 142-143.
   - Name of the two sampling locations instead of dates (Fig. 1) and replace "the sampling site" accordingly in l. 118.
   - Meaning of the Burger number (l. 147) and application to rotational internal waves.
   - Formula for M and V in l. 168.
   - Conversion from conductivity to salinity, as it is used in l. 248-249.

- Calculation of the salinity-dependent density in the model (l. 209-224). The salinity effects are included in Eq. (9), Eq. (10) and in l. 224, but it is unclear how the density is calculated from those quantities. Is the density calculated with TEOS-10 in the model as for field data (l. 130)?
- In-situ measurement of Secchi depth (value given in l. 228).
- Varying parameters between the simulations (sensitivity analysis), instead of fixed parameters in l. 227. The authors should also explain why the selected lake depth of 7 m (mean depth of the entire North Aral Sea, l. 392) is smaller than the depth of 11 m given in l. 227, which I assume is the mean lake depth of the Shevchenko Bay.
- Remote sensing data and determination of ice-on and ice-off dates. It is currently unclear in l. 271 how these dates were defined, but I assume that they come from the satellite images (Fig. 4)? The reference to satellite images is also missing in the description of the spatial variability of the ice cover in l. 276-281.
- Direction of the z-axis. It seems downward and positive in Eq. (1), Fig. 3b and l. 331, but upward and negative in Eqs. (2) and (7).

6. The figures are visually clear, but they lack some information, especially in the captions and in their interpretation.

Figure 2:
- Are the conductivity profiles corrected for temperature?
- A short description of the oxygen profiles should be added to the text.
- Why is oxygen decreasing near the surface in 2016?
- Why is there a bottom increase in Chl-a in 2016 and 2018?

Figure 3:
- The temperature color bar orientation in (a) is confusing because it is merged with the x-axis. I suggest orienting it vertically on the right of the plot.
- Are the transitions between lake seasons defined visually or based on $N^2$ or on the heat fluxes?
- How was the stratification threshold $N^2 > 2 \times 10^{-3}$ s$^{-2}$ selected and what does the thick black isoline provide?
- Indicate years of measurements in the caption.
- Would it be possible to add wind speed data from a meteo station or from ERA5 to explain internal-waves generation (or in Fig. 5) and to support the wind stress value given in l. 340?
- Could the ice cover period be indicated based on satellite images as in Fig. 4?
- Use lowercase letter for subpanels as in the other figures.

Figure 4: mention in the caption that the dot on satellite images shows the mooring location.

Figure 5:
- The frequency $\sigma$ does not match the dimensionless frequency defined in l. 158: I assume that it should be $\omega$ instead? Same in Fig. 8.
- Explicitly indicate that 4.5 d is a period and not a frequency, by moving the text away from the x-axis or relating it to the frequency as $\frac{2\pi}{\omega} = 4.5$ d.
- Use a different line style to indicate the Coriolis parameter $f$ as it can be confused with the $S(\omega)$ relationships, and mention this line in the caption.

- Indicate the meaning of the horizontal and vertical grey solid lines (e.g., value of the Burger number $S = 0.12$).

Figure 7:
- DO units are mg l$^{-1}$ on the y-axis but ppm in the caption and the text.
- The color of the DO saturation curve is orange rather than brown (adjust the caption).
- Mention in the caption the grey bar illustrating the ice cover and the meaning of thick lines (moving averages?).
- The depth of the DO sensor is 12 m in the caption, but it should be 10 m according to the methods (l. 122), which makes an important difference as the bottom low-oxygen layer starts below 12 m in Fig. 2a. The same depth of 12 m is also written in l. 484.
- Could you add temperature time series at the same depth as the DO data to highlight the effects of temperature oscillations on DO?
- There is a large peak at the end of the time series, with DO saturation reaching 130 %. Could it be due to mooring retrieval?

Figure 8:
- Indicate the periods of the two vertical lines associated with Kelvin waves, as the other periods. Why are there two peaks associated with Kelvin waves?
- The 12 h peak is not explained in the text (l. 381).

Figure 9:
- Indicate "mean lake depth" instead of "depth" in legend of (a), x-axis in (b) and caption.
- Indicate in the caption or on the y-axis of (b) the light extinction coefficient used in (a).
- Would it be possible to include the time series of modelled and observed surface temperatures in an additional panel? It would support l. 388-389.
- Mixing regimes are not explicitly indicated in (b) but discussed in l. 405-410. Would it be possible to add contour lines or another panel showing the two regimes as a function of mean lake depth and extinction coefficients? An option would be to use the number of mixing events during the year based on temperature differences of less than 1°C (l. 231).
- Changes in mixing regime mentioned in l. 397-398 should be moved later where the sensitivity analysis of Fig. 9b is described (l. 405).

Figure 10:
- The definition of $h_{mix}$ is unclear: is it the deepest mixed layer of the year? Where do the values of $h_{mix}/H$ come from?
- Better define the words used on the schematic. Is the stagnation layer the layer that remains stratified in summer? "Turbulence" is vague, why is it not present on the 3rd schematic?
- Are the dark and light grey shapes illustrating the location of the thermocline at two different times?
- Each schematic could be divided into 3 zones: fully mixed (white zone; O2>100%), wave-affected = polymictic (sloping sides) and permanently stratified (bottom, O2<100%).

7. The description of the annual stability (l. 330-347) based on Figure 6 could be inserted in the respective season-related subsections, instead of keeping it at the end of Sect. 3.2. and re-introducing the change in heat content in l. 331 (after having already described it in l. 292). Figure 6 could be combined with Figure 3.

8. The under-ice dynamics can be better described. The oscillations mentioned in l. 284-285 should be interpreted and a reference to Fig. 4 must be added. Explain where the under-ice convection (l. 287-288) is observed in Fig. 4 and mention the generation of internal waves as in l. 374. Could the temperature oscillations at 10 m before ice breakup also be due to differential heating? Correct l. 448-454 that state that under-ice convection is prevented in brackish lakes, which is inconsistent with the interpretation of Fig. 4.

9. The oxygen dynamics can also be better explained in Sect. 3.3. It would be useful to describe the oxygen stratification (less oxic bottom waters, Fig. 2a) at the beginning. The yearly high saturation mentioned in l. 350 might be specific to the sensor depth of 10 m, which seems to be above the bottom stratification under ice (Fig. 3a) and in summer (Fig. 2a). Hypoxia can occur in deeper layers and should be mentioned in l. 488. The oxygen drop after snowmelt (l. 358-361) is not explained: is it due to vertical mixing by under-ice convection?

10. References to previous studies should be added as follows:
    - l. 24-25: references for the list of endangered lakes.
    - l. 27-29: references about restoration measures.
    - l. 237-240: the first paragraph of the results is misleading as it starts with the salinity data that was already published in Andrulionis et al. (2022), it would be clearer to cite this source in l. 237 after "observational results". Is there a need to mention water sampling in l. 114-115 if this data is already included in Andrulionis et al. (2022)?
    - l. 209-217: the text is the same as in the model documentation (Mironov, 2005), which is not cited. Please rewrite these sentences instead of copying-pasting them.

In addition, please check the references in the bibliography: a doi link is sometimes included twice for the same reference and it is sometimes absent, capital letters are sometimes incorrectly used in the article titles.

**Minor comments:**

- Units have a different font than other letters in the text.
- Font style and size vary between figures.
- Some sentences could be more concise to improve readability (e.g., l. 5-7, l. 30-36, l. 286-289, l. 444-447).
- There are many occurrences of "North Aral". I would always use "Sea" after "Aral" to be consistent.
- l. 8: "annual" is a repetition of "year-long".
- l. 10: "cold restart" is unclear, please give more information or replace by "restauration".
- l. 13: ~4.5 days
- l. 16-17: dissolved matter (other than oxygen) and nutrients are not investigated in the study.
- l. 20 and l. 99: specify "1D modeling"
- l. 25: replace "threatening" by "causing" or "inducing"
- l. 29: replace "lake level replenishment" by "water replenishment" or "lake level increase"
- l. 63: remove "by this"
- l. 66: remove "conversely"
- l. 70: remove "there"
- l. 83: replace "to" by "for"
- l. 87: remove "in turn"
- l. 94, 95: no capital letter after (i) and (ii)
- l. 95 "periods of stagnation", l. 96 "stagnation phases", l. 101 "potential stagnation": why not "stratification periods"?
- l. 97: replace "below" by "in this study"
- l. 99: remove "climate scenarios" since it is already part of modelling
- l. 108,110,117: not same precision of coordinates
- l. 110-111: the verb is missing (e.g., "...were performed")
- l. 115: period missing at the end of the sentence
- l. 190: define "ODE"
- l. 225: define "ERA5"
- l. 256: repetition of "not"
- l. 273: "8 ‰" (space missing)
- l. 289: "referred TO as"
- l. 317: "Eq. (2)"
- l. 441: replace "preventing" by "delaying" since ice forms in winter
- l. 443: repetition of "the"
- l. 471: replace "turning the lake to polymictic" by "and could turn the lake to polymictic", otherwise it could be interpreted as if the lake was polymictic.
- l. 491: ppm instead of ppt

**References**

Mironov, D. V. (2005). Parameterization of lakes in numerical weather prediction. Part 1: Description of a lake model. *German Weather Service*, Offenbach am Main, Germany. https://www.cosmo-model.org/content/model/cosmo/misc/flake/docs/ParLak_Part1_a.pdf

---

## Author Response (AR1)

Authors' Response to Reviews of

**Consequences of the Aral Sea restoration for its present physical state: temperature, mixing, and oxygen regime**

G. Kirillin, T. Shatwell, A. Izhitskiy
Hydrology and Earth System Sciences,
* * *
RC: Reviewers' Comment,    AR: Authors' Response,    □ Manuscript Text

**1. Reviewer #1**

RC:  This study investigates the current mixing dynamics of the western basin of the North Aral Sea (Shevchenko Bay), 15 years after a dam was built as an attempt to save the North Aral Sea from desiccation. The study demonstrates from year-round measurements of thermal stratification and oxygen that a thin and low oxygen bottom stratified layer persists in summer as in dimictic lakes. However, inertial internal waves move the stratified layer around, causing the sloping sides of the Shevchenko Bay to be fully mixed several times a year as in polymictic lakes. The authors call this mixing regime "transitional" between dimictic and polymictic and show with 1D numerical simulations that an increase (decrease) in water level and light attenuation would bring the lake to a steady dimictic (polymictic) regime. Although the physical processes quantified by the study are well known, the fact that they are observed in a unique, evolving system makes the findings of the study very interesting and suitable for publication in HESS. The analysis that combines year-round temperature and oxygen measurements, CTD profiles, satellite images and 1D numerical modelling is comprehensive and robust. Yet, several statements are misleading or lack detailed explanation. Most of my comments aim at improving the clarity and readability of the manuscript.

AR:  We thank the Reviewer for the positive evaluation of our study and appreciate the recommendation for publication of our findings in HESS. While the physical processes considered in our study are indeed known, their interplay and the resulting mixing pattern has not been described before: In large lakes with the mixed layer thickness comparable to the mean lake depth, the basin-scale internal waves produce the effect of wandering hypolimnion with important consequences for near-bottom mixing and littoral-pelagic exchange. We are unaware of any previous studies discussing this mixing regime. We particularly appreciate the detailed comments and suggestions of the Reviewer, which helped improving our results presentation. We address them in our point-to-point replies and have incorporated the majority of the suggestions in the revised paper.

Specific comments

RC:  The location of the measurements is different from typical monitoring at the deepest point of the lake, which should be more emphasized through the manuscript. In fact, the data has been collected on the sloping sides of the western basin of the North Aral Sea (Shevchenko Bay; Fig. 1), which (i) might not be representative of the other basins of the North Aral Sea (including the deeper, central basin) and (ii) differs from measurements at the deepest point of the Shevchenko Bay. The authors should clearly state in the abstract, results, and discussion that their analysis focuses exclusively on the western lake basin. Additionally, they should elaborate on any potential differences or similarities

in mixing dynamics between the western and central basins. Knowing the presence of an east-west salinity gradient (l. 241) and a $\sim 7$ m deep sill between the two basins (Fig. 1), do the authors expect wind-driven or density-driven basin exchange that could have affected their measurements? The mooring misses the bottom of the water column because it was not deployed at the deepest point of the Shevchenko Bay. An additional mooring at the deepest point would have provided essential information on the full water column, including the persistence of the bottom stratified layer in summer. The authors should clearly indicate this limitation in the methods and that the frequent mixing events (polymictic-type) were only observed on the sloping sides of the basin.

AR: We fully agree with the Reviewer: an additional mooring in the center of the Shevchenko Bay and a couple of moorings in other parts of the North Aral would have allowed obtaining a more comprehensive quantitative picture on the lateral variability. Unfortunately, field investigations are inevitably constrained by available instrumentation and logistic issues. The fact that our mooring was positioned away from the lake center allowed us to capture the wave-driven lateral motions of the thermocline. The latter could have been missed if we chose the "typical" location at the deepest point of the lake, which is close the nodal point of the basin-scale waves in nearly-round Shevchenko Bay (cf. mid-panel of Fig. 10 in the ms). The lateral water exchange between the Shevchenko Bay is not expected to significantly affect the temperature, stratification, and waves pattern, because the upper 7 m of the water column remain well-mixed vertically (Fig. 3A), i.e. the water characteristics on both sides of the 7-m deep strait are nearly identical. We mentioned potential differences between different parts of the lake (Lines 504-507). These differences do not affect the core results of the study: The potential effect of the river inflow on the east-west salinity gradient is expected to be limited to the eastern shallow part of the lake (Fig. 1) and does not influence thermal stratification and vertical mixing in our observations. Taking into account the similar morphometries of the Shevchenko Bay and of the central part of the Aral Sea, the same wave pattern should dominate the mixing in the bulk of the Sea except shallow bays in the north and at the river estuary.

RC: It would help the reader to better explain the past, current and future mixing regimes of the Aral Sea. First, explicitly define "dimictic" and "polymictic" where the mixing regime is introduced in l. 61-67 of the introduction and remove the word "dimictic" in l. 43. The comparison between the natural (dimictic), pre-restoration (polymictic?) and current (transitional) mixing regimes of the Aral Sea should also be clearer by (i) providing information in the abstract about the past mixing regime (l. 18) before describing the current mixing regime in l. 10 and (ii) using the natural mixing regime as a reference point through the manuscript. The end of the abstract and the conclusion do not provide a clear answer regarding the preferred regime: is the goal to return to the natural, dimictic regime? The current, "intermediate" (l. 10) or "transitional" (l. 409, 496) mixing regime should be better explained as it could be incorrectly interpreted as a regime alternating between polymictic and dimictic temporally (i.e., between years), whereas this variation is spatial(i.e., frequent mixing on the sloping sides, persistent stratified layer at the deepest point). As explained in my first comment, the authors should provide hypotheses about the persistence of summer stratification at the deepest point. They could estimate the percentage of the Shevchenko Bay that remains stratified (based on a 2-3 m stratified layer, l. 474) instead of using the term "small" (l. 11, 475, 486). Regarding the future mixing regime, I suggest to explicitly mention the effects of an increase (or decrease) in water level and light attenuation instead of referring to "changes" or "shifts" which remain vague (for example in l. 20 and l. 96).

AR: We have added definitions of dimictic and polymictic regimes at Lines 66-70 and referred the reader to more details in (Kirillin and Shatwell, 2016). We prefer to avoid calling any mixing regime "natural" or to use it as a reference point. We know that the Aral Sea was dimictic as early as 1904 (Berg,

1908). There are no earlier observations on the vertical thermal structure, but paleolimnological studies suggest that the Aral Sea underwent strong lake level variations on centennial time scales (Boomer et al., 2000), which would have inevitably caused changes in the seasonal mixing regime. Therefore, we focus on the specific features of the present mixing regime, which is different from the commonly considered in conventional lake classification. Hence, there is no preferred seasonal mixing regime for the North Aral. The present regime appears to be advantageous, preventing extensive deep hypoxia typical for dimictic lakes. In turn, polymictic conditions may provoke massive algal blooms due to enhanced nutrients supply from the sediment. We discussed these scenarios in the ms (Line 501 onwards). One usage of the word "intermediate" and two usages of the word "transitional" are always explicitly explained in the context of the discussion and should not confuse the reader. Regarding the hypothesis on existence of the summer stratification in the deep parts of the North Aral: we report the evidence of it, based on the analysis in terms of basin-scale waves and supported by 1D modeling results. The existence of the deep stratification is the only possible source of the observed temperature and oxygen variations. We have explicitly mentioned and quantified the potential effects of water level and light attenuation on stratification by means of modeling (see Section 3.4). The consequences for the biogeochemistry can be discussed only qualitatively, what we did in the last paragraph of Discussion. The words "changes" and "shifts" are used in context, because both increase and decrease of corresponding factors affect seasonal stratification. The specific potential effects of an increase or decrease in water level and light attenuation are described in the last paragraph of revised Discussion. We have added an estimate of the stratified volume (about 7 %) to the revised Abstract and Discussion.

RC:     The information about the morphology of the Ara Sea and its history should be re-organized in a more logical way. Volume and surface areas given in the abstract (l. 2-3) and in the conclusion (l. 526-527) should be moved to the introduction where the Aral Sea is presented (e.g., l. 50-54). A few characteristics of the restoration project (e.g., construction of a dam) should be given as soon as the "Aral Sea restoration project" is mentioned in l. 37. The name of the river inflows and the dam must be given as soon as they are introduced in l. 40 and l. 48, respectively. The paragraph about the climate of the Aral Sea region (l. 84-88) should be moved earlier in the introduction, maybe just after the mixing regime has been introduced (l. 67). The importance of the North Aral Sea (l. 88-92) should also be stated earlier, where it is introduced in l. 40 for example. Differences between the Southern (hypersaline) and Northern (brackish) Aral Sea could also be developed in the introduction so that the comparison between the mixing dynamics of the Northern and Southern Aral Sea (l. 428-435) would be clearer. Is $T_{md} > T_f$ in the Southern Aral Sea?

AR:     We have reorganized the flow of introduction according to the Reviewer's suggestions where appropriate (see revised Introduction). The Southern Aral Sea is an interesting research object with extreme environmental conditions (Izhitskiy et al., 2016), but is out of scope of this study and is mentioned only as an asymptotic example of what could have happened to the North Aral Sea without restoration measures. Like all waters with salinity $\gtrsim$ 24.7 ppt, Southern Aral has $T_{md} = T_f$ (cf. the beginning of the paragraph cited by the Reviewer: "In contrast to ocean waters and hypersaline lakes. . .".

RC:     The assumption that salinity has negligible effects on stratification (l. 132-133) must be more justified and nuanced. Could the authors compare estimates of vertical bottom density gradient from conductivity profiles and from temperature profiles (e.g., June CTD profile) to support their assumption? Although salinity-driven stratification is negligible in summer, salinity effects might still play a role during turnover periods, when water temperature reaches the temperature of maximum density. It seems that an inverse thermal stratification occurred in mid-November (Figs. 3a, 4),

which would indicate a role of salinity in maintaining a (weak) stratification. Salt exclusion could also generate convection (ice growth) or stratification (ice melt) in winter. The model does not include this process (l. 201-203), but l. 448-454 state that salt exclusion is important in brackish lakes, which seems inconsistent. In addition, the model assumes constant salinity over time (l. 203), which differs from the continuous decrease in salinity observed after the dam construction (Fig. 2) and from the natural seasonal variability of salinity (l. 76-77). The authors should mention this model limitation and discuss the potential effects of a further decrease in salinity on the mixing dynamics: would a decrease in salinity reduce the thermal expansivity of water (Eq. 10) and the tendency of the lake to stratify?

AR: We agree with the Reviewer: the role of salinity on vertical stability of the water column cannot be completely ignored: short-term and/or localized events of salt stratification are not to be excluded. The three short-term ($< 1$ day) events of inverse stratification in November were rather driven by the combined effect of differential cooling between near-shore and central areas and wind surges. The inversions were immediately followed by strong convective mixing, which quickly eliminated the stratification and returned the water column to the perfectly mixed state. The free convection, reflected by high-frequency oscillations in the temperature records, would not develop if salinity contributed to the vertical stability. Herewith, the role of salinity gradients in the autumn overturn and the summer stratification is apparently negligible, as acknowledged by the Reviewer. The slight increase of the electric conductivity near bottom in the June CTD profile makes only a modest contribution to stability: the mean density ratio within the layer is around 15, i.e. temperature contribution to stability is 15 times higher than that of salinity. The vertical conductivity gradient in this layer is therefore a consequence of the temperature-driven stratification and subsequent accumulation of dissolved matter near the sediment surface. Vertical mixing under ice and near-bottom stratification during the spring overturn are indeed prone to be affected by salt stratification, and we discussed it throughout the ms. We have refined this discussion by adding the following text: "Our modeling projections, while reliably simulated the seasonal thermal stratification pattern, did not take these processes into account. Incorporating of the salinity effects of vertical salt gradients may improve short-term model predictions of the vertical stratification in brackish lakes, especially in the ice-covered period and during the spring overturn, when the density dependency on temperature vanishes and the near-bottom salinity gradients may prevent convective mixing at the water-sediment interface (Mironov et al., 2002)". Decrease of salinity to zero would reduce the thermal expansion coefficient by about 11 %; no remarkable effects on stratification on seasonal time scales should be expected.

RC: The methods section lacks the following information for better clarity:

- Maximal and mean depth of the North Aral Sea and the Shevchenko Bay.

- Value of the gravity acceleration in l. 127 (instead of l. 216).

- Formula of $g'$ and $h_{eq}$ in l. 142-143.

- Name of the two sampling locations instead of dates (Fig. 1) and replace "the sampling site" accordingly in l. 118.

- Meaning of the Burger number (l. 147) and application to rotational internal waves.

- Formula for M and V in l. 168.

- Conversion from conductivity to salinity, as it is used in l. 248-249.

- Calculation of the salinity-dependent density in the model (l. 209-224). The salinity effects are included in Eq. (9), Eq. (10) and in l. 224, but it is unclear how the density is calculated from those quantities. Is the density calculated with TEOS-10 in the model as for field data (l. 130)?

- In-situ measurement of Secchi depth (value given in l. 228).

- Varying parameters between the simulations (sensitivity analysis), instead of fixed parameters in l. 227. The authors should also explain why the selected lake depth of 7 m (mean depth of the entire North Aral Sea, l. 392) is smaller than the depth of 11 m given in l. 227, which I assume is the mean lake depth of the Shevchenko Bay.

- Remote sensing data and determination of ice-on and ice-off dates. It is currently unclear in l. 271 how these dates were defined, but I assume that they come from the satellite images (Fig. 4)? The reference to satellite images is also missing in the description of the spatial variability of the ice cover in l. 276-281.

- Direction of the z-axis. It seems downward and positive in Eq. (1), Fig. 3b and l. 331, but upward and negative in Eqs. (2) and (7).

AR:

- The information was added to the methods section: The bathymetry of the North Aral Sea can be estimated only approximately and is continuously varying within the seasonal hydrological cycle. The last bathymetric map of the Aral Sea was created in 1960ties (Bortnik and Chistyaeva, 1990). Extrapolating these data on the present satellite altimetry and using selected depth measurements from our surveys, the mean depth of the North Aral Sea varies in the range of 6-8 m with the maximum depths around 18-18 m. The mean depth of the Shevchenko Bay is 7–8 m, with maximum depths reaching up to 15-16 m in its central part.

- corrected

- We have restructured Eq. 3 so that the meaning of $g'$ and $h_{eq}$ is now evident for the reader.

- Figure 1 and the text have been modified accordingly

- The meaning of the Burger number is given by Eq. 4 and the accompanying text. Its application to rotational waves is evident from Eqs. 5-6. We have added a reference to (Gill, 1982) for further details.

- $M$ is the mass and $V$ is the volume. We have added the formulae to avoid any confusion.

- We have refined the text and added the reference to (Pawlowicz, 2008)

- The model uses the simplified equation of state in form of a 2-nd order polynomial (Eq. 8). As it is stated in the manuscript, the equation of state does not include the direct dependence of density on salinity, but is extended by the salinity effects on temperature of maximum density and on the thermal expansion coefficient. We thank for highlighting this point. The description of these effects in Methods was uncertain and is revised now

- A note on the Secchi measurements was added

- We have corrected the the model setup description (L227 of the original manuscript). See also our response on the mean and maximum depths above.

- We have added "satellite-derived" with a reference to Fig. 4 to the text

- We have changed the integration limits in Eqs.(2), (7) for consistency

RC: Figure 2:

- Are the conductivity profiles corrected for temperature?

- A short description of the oxygen profiles should be added to the text.

- Why is oxygen decreasing near the surface in 2016?

- Why is there a bottom increase in Chl-a in 2016 and 2018?

AR: We have revised Fig. 2: the in situ conductivity profiles were replaced by specific conductivity C25 and the corresponding description was refined; A short description of oxygen profiles is added. Near-surface decrease of DO concentrations in summer might be caused e.g. by photoinhibition of primary production by strong solar radiation. Summer time series of DO with sub-daily resolution are required for a more distinct answer. The Chl-a peaks at the bottom are common in lake measurements (G. Kirillin, pers. comm.) and may be caused by accumulation of fluorescent matter at the sediment surface.

RC: Figure 3:

- The temperature color bar orientation in (a) is confusing because it is merged with the x-axis. I suggest orienting it vertically on the right of the plot.

- Are the transitions between lake seasons defined visually or based on $N^2$ or on the heat fluxes?

- How was the stratification threshold $N^2 > 2 \times 10^{-3}$ s$^{-2}$ selected and what does the thick black isoline provide?

- Indicate years of measurements in the caption.

- Would it be possible to add wind speed data from a meteostation or from ERA5 to explain internal-waves generation (or in Fig. 5) and to support the wind stress value given in l. 340?

- Could the ice cover period be indicated based on satellite images as in Fig. 4?

- Use lowercase letter for subpanels as in the other figures.

AR: The horizontal colorbar was chosen to keep the x-axes of both panels aligned without affecting the aspect ratio of the figure. We have separated the colorbar from the second panel to improve readability. The end of spring mixing period and the start of the fall overturn were determined visually. The beginning and the end of the winter period coincide with the ice-covered period from satellite measurements. Meteorological data would make the figure overloaded with information without adding any crucial information. As it is stated in the caption, thick black isoline is the boundary of the bottom stratified layer defined as $N^2 > 2 \times 10^{-3}$ s$^{-2}$. The value is the approximate

maximum of $N^2$ over the period of measurements. Years of measurements have been added to the caption.

RC: Figure 4: mention in the caption that the dot on satellite images shows the mooring location.

AR: Done.

RC: Figure 5:

- The frequency $\sigma$ does not match the dimensionless frequency defined in l. 158: I assume that it should be $\omega$ instead? Same in Fig. 8.

- Explicitly indicate that 4.5 d is a period and not a frequency, by moving the text away from the x-axis or relating it to the frequency as $2\pi/\omega = 4.5$ d.

- Use a different line style to indicate the Coriolis parameter $f$ as it can be confused with the $S(\omega)$ relationships, and mention this line in the caption.

- Indicate the meaning of the horizontal and vertical grey solid lines (e.g., value of the Burger number $S = 0.12$).

AR: To avoid confusion with the angular speed of the earth rotation, $\omega$ is replaced by $\sigma$ and $\sigma$ is replaced by $\varsigma$. We moved information on wave periods to the figure legend. The legend is extended to explain the meaning of the inertial frequency and the energy peaks.

RC: Figure 7:

- DO units are mg l-1 on the y-axis but ppm in the caption and the text.

- The color of the DO saturation curve is orange rather than brown (adjust the caption).

- Mention in the caption the grey bar illustrating the ice cover and the meaning of thick lines (moving averages?).

- The depth of the DO sensor is 12 m in the caption, but it should be 10 m according to the methods (l. 122), which makes an important difference as the bottom low-oxygen layer starts below 12 m in Fig. 2a. The same depth of 12 m is also written in l. 484.

- Could you add temperature time series at the same depth as the DO data to highlight the effects of temperature oscillations on DO?

- There is a large peak at the end of the time series, with DO saturation reaching 130 %. Could it be due to mooring retrieval?

AR: We made the units uniform throughout the text. The caption is extended as suggested. The deployment depth in Methods is corrected. The DO peak in smoothed DO values could be produced by the endpoint effect of moving average. Removed in the revised figure. Near-bottom temperature time series added.

RC: Figure 8:

- Indicate the periods of the two vertical lines associated with Kelvin waves, as the other periods. Why are there two peaks associated with Kelvin waves?

- The 12 h peak is not explained in the text (l. 381).

AR: In the figure caption, we have indicated the Kelvin wave period matching the same period in the temperature oscillations. The 12-h peak is also commented in the caption as a result of primary production, which is not purely diurnal but ceases during the nighttime.

RC: Figure 9:

- Indicate "mean lake depth" instead of "depth" in legend of (a), x-axis in (b) and caption.

- Indicate in the caption or on the y-axis of (b) the light extinction coefficient used in (a).

- Would it be possible to include the time series of modelled and observed surface temperatures in an additional panel? It would support l. 388-389.

- Mixing regimes are not explicitly indicated in (b) but discussed in l. 405-410. Would it be possible to add contour lines or another panel showing the two regimes as a function of mean lake depth and extinction coefficients? An option would be to use the number of mixing events during the year based on temperature differences of less than 1°C (l. 231).

- Changes in mixing regime mentioned in l. 397-398 should be moved later where the sensitivity analysis of Fig. 9b is described (l. 405).

AR:

- "mean lake depth" has been explicitly indicated in the revised caption.

- The light extinction coefficient and the mean lake depth used in Panel A have been added to the figure caption.

- The RMSE for the observed surface temperatures is given in the text. 1D models, including FLake, are known to simulate surface temperatures well. We find an additional panel superfluous.

- We have added a rough definition of mixing regime transition to the figure caption.

- The changes in mixing regime mentioned in l. 397-398 are the prerequisite for the subsequent sensitivity analysis.

RC: Figure 10:

- The definition of $h_{mix}$ is unclear: is it the deepest mixed layer of the year? Where do the values of $h_{mix}/H$ come from?

- Better define the words used on the schematic. Is the stagnation layer the layer that remains stratified in summer? "Turbulence" is vague, why is it not present on the 3rd schematic?

- Are the dark and light grey shapes illustrating the location of the thermocline at two different times?

- Each schematic could be divided into 3 zones: fully mixed (white zone; O2>100%), wave-affected = polymictic (sloping sides) and permanently stratified (bottom, O2<100%).

AR:

- The definition of $h_{mix}$ as the summer-mean thickness of the upper mixed layer has been added to the caption.

- The schematic has been revised.

- yes.

- The suggestion is appreciated.

RC: The description of the annual stability (l. 330-347) based on Figure 6 could be inserted in the respective season-related subsections, instead of keeping it at the end of Sect. 3.2. and re-introducing the change in heat content in l. 331 (after having already described it in l. 292). Figure 6 could be combined with Figure 3.

AR: We prefer to keep the original manuscript flow: first, describe the phenomenological features of the intermittent stratification, second, discuss the driving mechanisms of the observed phenomena.

RC: The under-ice dynamics can be better described. The oscillations mentioned in l. 284-285 should be interpreted and a reference to Fig. 4 must be added. Explain where the under-ice convection (l. 287-288) is observed in Fig. 4 and mention the generation of internal waves as in l. 374. Could the temperature oscillations at 10 m before ice breakup also be due to differential heating? Correct l. 448-454 that state that under-ice convection is prevented in brackish lakes, which is inconsistent with the interpretation of Fig. 4.

AR: We agree, the under-ice dynamics in the Aral Sea definitely deserves a closer attention. The under-ice circulation and mixing in brackish lakes remain poorly investigated to date. Their analysis is a subject of a separate study and cannot be considered in details in the framework of the present paper. The wave origin of oscillations are mentioned in the text and a reference to Fig. 4 has been added to the revised ms. The origin and characteristics of these waves are subject of dedicated research and are not further discussed in the present study. Effects of differential heating can be neither excluded nor confirmed by these data. We refined the text to emphasize that vertical salt gradients prevent thermally-driven convection.

RC: The oxygen dynamics can also be better explained in Sect. 3.3. It would be useful to describe the oxygen stratification (less oxic bottom waters, Fig. 2a) at the beginning. The yearly high saturation mentioned in l. 350 might be specific to the sensor depth of 10 m, which seems to be above the bottom stratification under ice (Fig. 3a) and in summer (Fig. 2a). Hypoxia can occur in deeper layers and should be mentioned in l. 488. The oxygen drop after snowmelt (l. 358-361) is not explained: is it due to vertical mixing by under-ice convection?

AR: The oxygen sensor was positioned near the bottom, at water depth of 11.7 m. We have corrected the information on the sensor depth in the text (see also our replies above). We mentioned in Discussion

that localized hypoxia can still be found in deep parts of the ice-covered lake. The oxygen drop after snowmelt might be caused by entrainment of under-saturated water from the deep parts of the lake. Quantification of the mechanism requires requires additional spatially-resolved observations. We have added a note on this to the revised ms.

RC: References to previous studies should be added as follows:

- l. 24-25: references for the list of endangered lakes.

- l. 27-29: references about restoration measures.

- l. 237-240: the first paragraph of the results is misleading as it starts with the salinity data that was already published in Andrulionis et al. (2022), it would be clearer to cite this source in l. 237 after "observational results". Is there a need to mention water sampling in l. 114-115 if this data is already included in Andrulionis et al. (2022)?

- l. 209-217: the text is the same as in the model documentation (Mironov, 2005), which is not cited.

AR: References for the global decline in endorehic basins Wang et al. (2018) and about restoration measures Yapiyev et al. (2017) were added. We believe that there is a need to mention water sampling in Methods to let the reader know that CTD-measurements and thermistor chain installation/take off were done directly along with water sampling to determine the exact value of salinity. Meanwhile, we left the detailed description of laboratory procedures in the separate article (Andrulionis et al., 2022) directly citing the results.

RC: In addition, please check the references in the bibliography: a doi link is sometimes included twice for the same reference and it is sometimes absent, capital letters are sometimes incorrectly used in the article titles.

AR: Apparently, there are some issues with the Copernicus Bibtex template. We made our best to clean up the bibliography. Remaining issues can be fixed during copyediting.

RC: : Minor comments

- Units have a different font than other letters in the text.

- Font style and size vary between figures.

- Some sentences could be more concise to improve readability (e.g., l. 5-7, l. 30-36, l. 286- 289, l. 444-447).

- 'There are many occurrences of "North Aral". I would always use "Sea" after "Aral" to be consistent

- l. 8: "annual" is a repetition of "year-long".

- l. 10: "cold restart" is unclear, please give more information or replace by "restauration".

- l. 13: $\tilde{4}.5$ days

- l. 16-17: dissolved matter (other than oxygen) and nutrients are not investigated in the study.

- l. 20 and l. 99: specify "1D modeling"

- l. 25: replace "threatening" by "causing" or "inducing"

- l. 29: replace "lake level replenishment" by "water replenishment" or "lake level increase"

- l. 63: remove "by this"

- l. 66: remove "conversely"

- l. 70: remove "there"

- l. 83: replace "to" by "for"

- l. 87: remove "in turn"

- l. 94, 95: no capital letter after (i) and (ii)

- l. 95 "periods of stagnation", l. 96 "stagnation phases", l. 101 "potential stagnation": why not "stratification periods"?

- l. 97: replace "below" by "in this study"

- l. 99: remove "climate scenarios" since it is already part of modelling

- l. 108,110,117: not same precision of coordinates

- l. 110-111: the verb is missing (e.g., "...were performed")

- l. 115: period missing at the end of the sentence

- l. 190: define "ODE"

- l. 225: define the response 'ERA5'

- l. 256: repetition of "not"

- l. 273: "8 ‰" (space missing)

- l. 289: "referred TO as"

- l. 317: "Eq. (2)"

- l. 441: replace "preventing" by "delaying" since ice forms in winter

- l. 443: repetition of "the"

- l. 471: replace "turning the lake to polymictic" by "and could turn the lake to polymictic", otherwise it could be interpreted as if the lake was polymictic.

- l. 491: ppm instead of ppt

AR:

- This is a copyediting issue. We used the standard "unit" latex package

- We did our best to make the figure formatting uniform

- We have edited the sentences.

- We have changed all appearances to "The North Aral Sea".

- l. 8: "annual" is removed

- l. 10: fixed

- l. 13: fixed

- l. 16-17: The sentence refers to the discussion/conclusions part. We retain it in the abstract as an important outlook from our results.

- l. 20 and l. 99: There is no need for model details at these points.

- l. 25: changed to "causing"

- l. 29: changed to "water replenishment"

- l. 63: changed to "thereby"

- l. 66: removed

- l. 70: removed

- l. 83: replaced

- l. 87: removed

- l. 94, 95: fixed

- l. 95 "stagnation" and "stratification" are (almost) interchangeable. The former term has a stronger accent on biochemical consequences of stratification and is used accordingly.

- l. 97: "below" is properly used.

- l. 99: "climate scenarios" have been retained.

- l. 108,110,117: fixed

- l. 110-111: the verb is "included"

- l. 115: fixed

- l. 190: done

- l. 225: done

- l. 256: fixed

- l. 273: corrected

- l. 289: corrected

- l. 317: corrected

- l. 441: replace "preventing" by "delaying" since ice forms in winter

- l. 443: corrected

- l. 471: done

- l. 491: fixed

**2. Reviewer #2**

RC:   The manuscript analyses the thermal and oxygen dynamics in the North Aral Sea after its artificial isolation from the desiccating old Aral Sea. A combination of intensive field campaigns and prolonged monitoring in the Shevchenko Bay allows the authors to obtain a good understanding of the current dynamics. The use of a one-dimensional model supports the analysis of some scenarios. I enjoyed reading the manuscript and I believe that it is a substantial piece of work that will make an important contribution to assessing the impact of the engineering works aimed at preserving the quality and of water and ecosystems of what remains of the old Aral Sea. I had only a few minor suggestions for improving the clarity of some of the text and figures, but after reading the very detailed comments of the Anonymous Referee #1, I feel that they do not need to be listed here. I convincingly support the publication of this manuscript.

AR:   We thank the Reviewer for the positive evaluation of our study and highly appreciate the explicit recommendation for publication. We hope that the presentation clarity has been improved in the revised version.

**3. Reviewer #1, additional comment**

RC:   I have an additional comment about the DO-temperature relationship. As mentioned in my previous comments about Figure 7, it would be very helpful to have temperature and DO time series at the same depth shown on the same figure and to discuss the correlation between the two time series in Sect. 3.3. I have tried to compare the two time series by combining Fig. 7 with Fig. 5A for the summer period (see the figure below). I do not understand why the DO concentration and saturation are both increasing during stratified periods (blue area on the figure) and decreasing during mixing periods (yellow area). I thought it was the opposite: stratified periods should be associated with the presence of a bottom low-oxygen layer as explained in l. 365-367 and mixing periods should bring more oxygen. Is it because I have incorrectly linked the two figures (e.g., the x-axis ticks in Fig. 7 do not correspond to the first day of the month as in Fig. 5A) or is there a physical explanation?

AR:   DO concentrations generally follow the wave-driven temperature pattern (see the attached Figure and the Fig. 8 of the revised manuscript): The concentrations drop at the beginning of stratification events indicating upwelling of oxygen-poor hypolimnetic waters. The DO dynamic in the internal wave shoaling zone is apparently more complex than that of temperature and affected by other processes such as mixing due to wave breaking, shear-induced water-sediment gas exchange, primary production and respiration. These processes are worth of a dedicated study on fine-scale DO dynamics in the littoral and their quantification requires detailed field observations. We have essentially revised Section 3.3 dedicated to the oxygen dynamics.

[Figure]

References

Berg, L. (1908). Л. Берг: Аральское море. Научные результаты Аральской экспедиции. Изв. Туркест. отд. Импер. Рус. геогр. о-ва.—Петербург.

Boomer, I., Aladin, N., Plotnikov, I., and Whatley, R. (2000). The palaeolimnology of the Aral Sea: a review. Quaternary Science Reviews, 19(13):1259–1278.

Bortnik, V. and Chistyaeva, S. (1990). Hydrometeorology and Hydrochemistry of the USSR Seas. Volume VII. The Aral Sea. Gidrometeoizdat, Leningrad, USSR.

Gill, A. E. (1982). Atmosphere-ocean dynamics. Academic press.

Izhitskiy, A., Zavialov, P., Sapozhnikov, P., Kirillin, G., Grossart, H., Kalinina, O., Zalota, A., Goncharenko, I., and Kurbaniyazov, A. (2016). Present state of the Aral Sea: diverging physical and biological characteristics of the residual basins. Scientific reports, 6(1):1–9.

Kirillin, G. and Shatwell, T. (2016). Generalized scaling of seasonal thermal stratification in lakes. Earth-Science Reviews, 161:179–190.

Mironov, D., Terzhevik, A., Kirillin, G., Jonas, T., Malm, J., and Farmer, D. (2002). Radiatively driven convection in ice-covered lakes: Observations, scaling, and a mixed layer model. Journal of Geophysical Research: Oceans, 107(C4).

Pawlowicz, R. (2008). Calculating the conductivity of natural waters. Limnology and Oceanography: Methods, 6(9):489–501.

Wang, J., Song, C., Reager, J. T., Yao, F., Famiglietti, J. S., Sheng, Y., MacDonald, G. M., Brun, F., Schmied, H. M., Marston, R. A., et al. (2018). Recent global decline in endorheic basin water storages. Nature geoscience, 11(12):926–932.

Yapiyev, V., Sagintayev, Z., Inglezakis, V. J., Samarkhanov, K., and Verhoef, A. (2017). Essentials of endorheic basins and lakes: A review in the context of current and future water resource management and mitigation activities in Central Asia. Water, 9(10).